# Engineering oxygen nonbonding states in high entropy hydroxides for scalable water oxidation

Fangqing Wang[1] ✉, Liu Feng[2], Mingwei Zhang[1,2] & Hailin Cong [1,3] ✉

The lattice oxygen oxidation mechanism typically requires the removal of electrons from the metal-oxygen band, which may cause structural instability due to a decrease in the metal-oxygen bond order. To address this challenge, we introduce low-valence, non-catalytically active Na to construct oxygen non-bonding bands on high-entropy hydroxides, allowing electrons to be removed from the oxygen non-bonding band rather than the metal-oxygen bonds, thereby improving the stability of the catalyst. Na doped high-entropy layered double hydroxide (Na-HE LDH) with a low overpotential of 176 mV@10 mA cm$^{-2}$ under alkaline conditions. Furthermore, the Pt/C||Na-HE LDH electrode pair operates continuously for 2000 h at ~500 mA cm$^{-2}$ in an anion-exchange membrane electrolyzer (30 wt% KOH, 60 °C). In-situ spectroscopic and density functional theory calculations identify that the introduction of Na facilitates the formation of oxygen non-bonding band thereby mitigating structural instability. This study offers a strategy for designing efficient and stable lattice oxygen catalysts and provides valuable insights for developing catalysts capable of withstanding the rigorous demands of industrial hydrogen production environments.

Oxygen evolution reaction (OER) involves a multi-electron transfer process with slow kinetic reaction, which is the key factor limiting the efficiency of the large-scale industrial application of water electrolysis[1-5]. Therefore, the development of highly active, durable and low-cost non-precious metal-based oxygenation electrocatalysts and the understanding of the intrinsic mechanism of the OER reaction are key to enhance the kinetics of the oxygenation reaction. Depending on the nature of the electronic structure in the catalyst, the active site can be switched from the metal center to the lattice oxygen center, thus switching the OER mechanism from adsorbate evolution mechanism (AEM) to lattice oxygen oxidation mechanism (LOM)[6-9] and greatly improving the OER kinetics[10,11].

Recently, a variety of oxides, hydroxides and hydroxyl oxides, such as NaxMn$_3$O$_7$[12], Fe/F−CoO[7], MoNiFe (oxy)hydroxide[11], have been successfully activated lattice oxygen by strategies such as defect modulation, construction of heterostructures and single-atom loading. However, although the above catalysts break through the limitation of the theoretical potential of AEM by being activated lattice oxygen and thus following the LOM pathway to obtain the better intrinsic activities, the LOM-based catalysts still suffer from the problem of poor stability, especially when OER is carried out at industrialized high current densities[13]. The underlying reasons are: (1) the conventional LOM mechanism-based catalysts undergo surface remodeling during the OER reaction, during which a large amount of surface active metal exists to leach out or polarize into a low-activity second phase, leading to catalyst deactivation. (2) The participation of lattice oxygen in the catalyst in the reaction generates causes a large number of oxygen vacancies, leading to the collapse of the bulk phase structure and phase transition, resulting in poor catalyst durability[13]. Therefore, the adoption of lattice oxygen activation strategies to

[1]School of Materials Science and Engineering, Shandong University of Technology, Zibo, PR China. [2]Analytical and Testing Center, Shandong University of Technology, Zibo, PR China. [3]College of Chemistry, Chemical Engineering and Materials Science, Zaozhuang University, Zaozhuang, China. ✉e-mail: wfq970111@163.com; conghailin@sdut.edu.cn

effectively avoid or inhibit the leaching and segregation of metal ions, as well as the collapse and phase transition of the bulk phase structure are the key to the development of LOM-based OER electrocatalysts with high activity and high stability[14,15].

Usually, the metal and oxygen orbitals of metal (oxygen) hydroxides hybridize to form (M−O) bonding bands with oxygen features and (M−O)* antibonding bands with metal features[16,17]. For late transition metal (oxygen) hydroxides[17], strong d-d Coulomb interactions split the (M−O)* antibonding band into an empty upper Hubbard band (UHB) and a full lower Hubbard band (LHB)[18]. In previous LOM catalysts, it has been common to lower the LHB so that the LHB enters the (M−O) bonding band[19]. In this way, the removed electrons have to come mainly from the (M−O) bonding band, which may lead to structural instability due to the reduced M−O bond order[18]. From the point of view of structural stability, triggering lattice oxygen oxidation in oxyhydroxide-based electrocatalysts requires ensuring the following two conditions: (1) Introduction of a buffer band from the $O_{NB}$ state, even if the electrons are removed from the $O_{NB}$ rather than the (M−O)[16]. (2) A high covalency of the M−O bonding to ensure that the electrons are removed from the $O_{NB}$ rather than from the LHB[17,18]. Therefore, we use stable high-entropy (oxygen) hydroxides as matrix materials, as their surface active metal leaching and second-phase segregation are inhibited and they maintain good catalytic stability during the reconstruction process[20–22], and we introduce low-valent and non-catalytically active Na into them to form the $O_{NB}$ state, while increasing the M−O covalency (Fig. 1). This class of materials also provides a disordered orbital environment favorable for the formation and activation of $O_{NB}$[23–25].

Herein, we report a low valence inactive main group Na ion doped high entropy hydroxide (Na-HE LDH) to achieve lattice oxygen activation for OER. advanced in situ spectroscopy shows that low valence Na ions are doped at the Fe sites simultaneously generating oxygen vacancies in situ. A combination of in-situ XAFS spectroscopy, $^{18}O$ isotope labeled mass spectra and density functional theory (DFT) calculations showed that doping of low-valent Na ions generates oxygen non-bonding states to hybridized adjacent oxygen vacancies to remove electrons, which shifts the OER pathway of HE LDH from AEM to LOM, lowering the energy barriers and enhancing intrinsic activity. The overpotential of Na-HE LDH catalysts at 10 mA cm⁻² with an overpotential of only 176 mV. More importantly, the Na-HE LDH could be operated continuously for 2000 h under harsh industrial-grade conditions (30 wt% KOH, 60 °C and ~500 mA cm⁻²), showing strong potential for industrial applications. In this study, the OER pathway of HE LDH from conventional AEM to LOM was modulated by introducing $O_{NB}$, which opens a way to regulate lattice oxygen in high-entropy LDH.

## Results
### Synthesis and structural characterizations of catalysts
Main-group Na ion-doped high entropy layered double hydroxides (Na-HE LDH) were synthesized by a simple one-step hydrothermal method, as shown in Fig. 2a. If not otherwise stated, all of the following Na-HE LDH refer to $Na_{0.045}$-HE LDH, this is due to its good OER properties, see Supplementary note 1, Fig. S1 and Table S1 for details. The X-ray diffraction (XRD) patterns showed that the HE LDH diffraction peaks were consistent with PDF#50-0235 a hexagonal hydrotalcite structure (Fig. S2). In addition, scanning electron microscopy (SEM) and transmission electron microscopy (TEM) images (Fig. S3 and Fig. S4a) observed HE LDH as ortho-hexagonal nanosheet morphology with smooth surfaces and sharp edges. High-resolution TEM (HRTEM, Fig. S4b) showed 2.593 Å crystallite spacing pairs consistent with the (101) crystallite surface of HE LDH. Selected-area electron diffraction (SAED, Fig. S4c) also confirms the single-crystal nature of HE LDH. The elemental mapping (Fig. S4d) shows that the Mn, Fe, Co, Ni, and Cu are uniformly distributed on the nanosheets and the atomic ratios of Mn, Fe, Co, Ni and Cu to all metals are 4.31%, 3.88%, 4.16%, 3.54% and 3.38%, respectively (Table S2). The above results indicate the successful preparation of high-entropy HE LDH.

After the introduction of Na ions, the crystal structure was consistent with that of the pristine HE LDH (XRD, Fig. 2b), and no additional diffraction peaks appeared. Raman spectra (Fig. 2c) show two representative bands at 400 - 600 cm⁻¹ belonging to the $E_g$ bending mode of δ(Ni−O) and $A_{1g}$ stretching mode of ν(Ni−O) in the LDH structure, respectively. The SEM and TEM images (Fig. S5 and Fig. 2d) demonstrated that the Na-HE LDH still retained the clean ortho-hexagonal nanosheet structure, and no obvious attachment of metal or metal oxide particles was observed. HRTEM (Fig. 2e) shows that the crystal spacing of the (101) facet in Na-HE LDH is 2.672 Å, which is slightly enlarged compared to the pristine HE LDH (2.593 Å), which is caused by the atomic radius of Na being larger than that of other metals. It is also observed that the Na-HE LDH are all crystalline structures with no amorphous layer attached (Fig. S6). In addition, selected area electron diffraction regions collected along the [511] crystal band axis also shows the single-crystal nature of the Na-HE LDH (Fig. 2f), displaying crystal planes that are also in general agreement with the XRD results. Atomic force microscopy (AFM) results showed that the thickness of Na-HE LDH nanosheets was 4.99 nm (Fig. 2g). According to previous studies, the thickness of monolayer LDH is about 0.76 nm[26]. Additionally, the layer spacing of Na-HE LDH was calculated to be 0.43 nm based on the diffraction peaks of the crystalline plane of (003) in XRD (calculated by Bragg's formula)[27]. So, the prepared Na-HE LDH was calculated as 4 layers, whereas pristine HE LDH is 5 layers (Fig. S7). This reduction in layer number may be related to the presence of Na ions, which could potentially modulate the interlayer interactions and stacking behavior by affecting the hydrogen-bonding network or local electrostatic environment[26,28]. EDS elemental mapping showed that Mn, Fe, Co, Ni, Cu, O and Na was uniformly dispersed in HE LDH nanosheets (Fig. 2h and Fig. S8), where, Na was doped with 0.19 at% (Table S2).

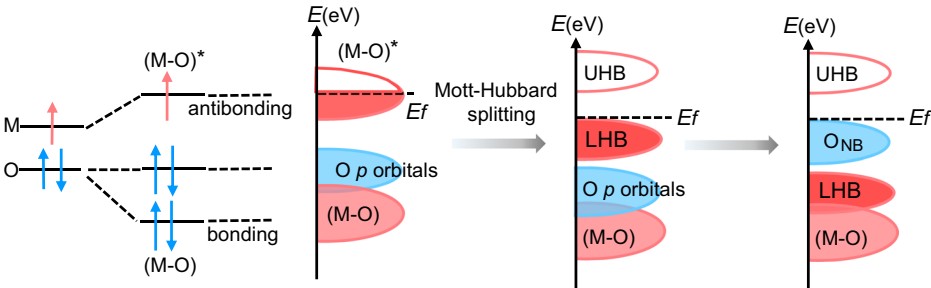

**Fig. 1 | Formation of $O_{NB}$ in Na-HE LDH.** Schematic diagram of the formation of $O_{NB}$ extrapolated from the molecular orbital energy diagram of Na-doped HE LDH. $E_f$ represents the Fermi level.

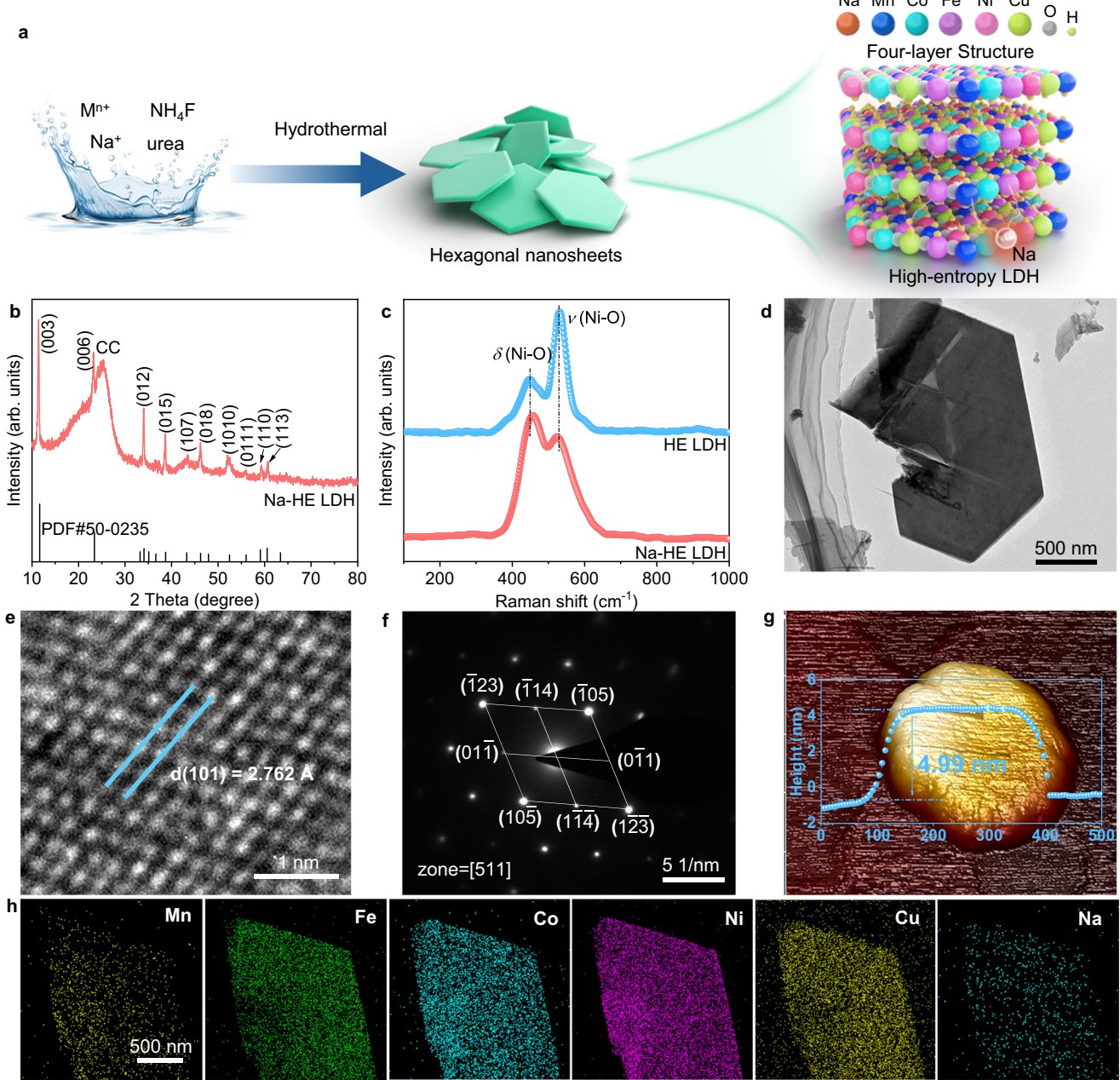

**Fig. 2 | Structural characterizations of Na-HE LDH. a** Schematic diagram of the synthesis of Na-HE LDH. **b** XRD. **c** Raman spectra of Na-HE LDH and HE LDH. **d** TEM. **e** HRTEM. **f** SAED and **g** AFM image of Na-HE LDH (inset is height profile of nanosheets). **h** EDS-Mapping of Na-HE LDH. Source data are provided as a Source Data file.

## Fine structures of catalyst materials

The elemental composition and chemical structure of the catalysts were analyzed by X-ray photoelectron spectroscopy (XPS). The presence of distinct characteristic peaks belonging to the Na-O bond located at 1071.75 eV in the Na 1$s$ spectrum of Na-HE LDH further confirms the successful doping of Na (Fig. 3a)[29]. Here we have to confirm the doping site of Na ions. We calculated the substitution formation energies of Na doping at each site by DFT (Fig. S9) and found that thermodynamically, Na prefers to occupy the position of Fe (Fe: −3.35 eV, Co: −2.88 eV, Ni: −3.23 eV, Mn: −2.68 eV, Cu: −2.65 eV). To further reveal why Na tends to occupy the Fe site, the metal vacancy formation energies of each site were calculated (Fig. S10), and the results show that among the many sites, Fe has a lower vacancy formation energy (Fe: −9.08 eV, Co: −8.78 eV, Ni: −8.89 eV, Mn: −8.46 eV, Cu: −8.77 eV), and Fe is more prone to break bonds and form vacancies, which provides an anchor site for Na. High-resolution XPS

spectroscopy showed that the binding energies of the 3d transitional elements in Na-HE LDH were shifted to higher energies compared to the pristine HE LDH (Fig. S11), indicates that the chemical state of the 3d transitional elements in Na-HE LDH is increased. This observed increase in the chemical state upon Na incorporation suggests a redistribution of d-electron occupancy, which may strengthen the M−O covalency by enhancing orbital overlap between metal 3$d$ and oxygen 2$p$ orbitals. This electronic modulation is expected to facilitate lattice oxygen participation, thereby favoring the LOM pathway over the conventional AEM during the OER process[30]. After Na doping, the XPS spectra of O 1$s$ are shifted towards lower binding energies, while all transition metal 2$p$ orbitals are shifted towards higher binding energies, indicating that the introduction of Na induces strong electronic interactions between the transition metal and oxygen. Additionally, from the XPS spectrum of O 1$s$ (Fig. 3b) found the presence of oxygen vacancies ($V_O$, 531.0 eV), indicating the presence of $V_O$ in the

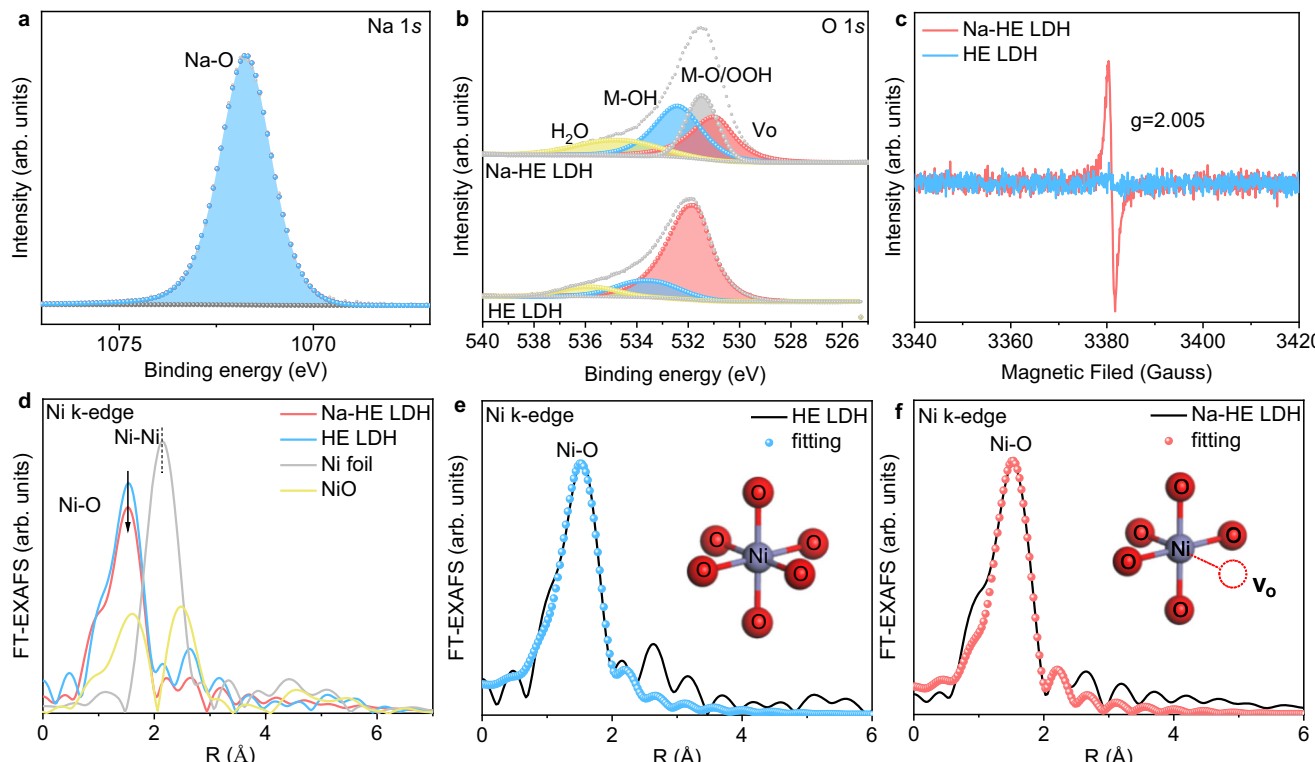

**Fig. 3 | Analysis of the electronic and coordination structure of Na-HE LDH.** **a** XPS spectra of Na 1$s$. **b** O 1$s$ of Na-HE LDH. **c** EPR spectra of Na-HE LDH and HE LDH. Signal at $g$ = 2.005 proves the presence of Vo in Na-HE LDH. **d** Ni k-edge FT-EXAFS spectra of Na-HE LDH, HE LDH, Ni foil and NiO, R is the radial distance. FT-EXAFS fitting curve of **e** HE LDH and **f** Na-HE LDH at Ni k-edge, the insert showing the fitting model. Source data are provided as a Source Data file.

oxygenated Na-HE LDH, a conclusion also supported by the EPR results (Fig. 3c). The valence changes of metal elements in the catalyst were further revealed by X-ray absorption near edge structure (XANES). Compared with pristine HE LDH, the absorption edge of the 3d transition metal in Na-HE LDH is located at a high-energy position with a higher oxidation state (Fig. S12). Based on linear extrapolation (Fig. S13), the valence states of Mn, Fe, Co, Ni and Cu in Na-HE LDH are calculated to be +2.49, +2.67, +2.42, +2.77 and +3.22, which is higher than that of Mn ( + 2.38), Fe ( + 2.57), Co ( + 2.29), Ni ( + 2.71), Cu ( + 2.81) in HE LDH, which is in agreement with the XPS results. To further reveal the changes in metal valence and M−O bond covalency, O K-edge soft X-ray absorption spectroscopy (sXAS) measurements of HE LDH and Na-HE LDH were carried out using the total electron yield (TEY) mode. sXAS spectra (Fig. S14) of the O K-edge of HE LDH and Na-HE LDH show distinct peaks at 534.5 and 540.5 eV, which are attributed to O 2$p$-M 3$d$ hybridization and O 2$p$-M 4$sp$ hybridization with, respectively[31,32]. It is noteworthy that the shift of the pre-edge peak to lower energies in Na-HE LDH is associated with an increase in the valence of the 3d element[6,33]. Moreover, Na-HE LDH exhibits a stronger pre-edge peak (534.5 eV) compared to HE LDH, which indicates enhanced metal-oxygen bond covalency[7,34].

Extended X-ray absorption fine structure (FT-EXAFS) was used to resolve the local coordination structure of Na-HE LDH. The peak intensities of Mn-O (1.64 Å), Co-O (1.60 Å), Fe−O bonds (1.51 Å) and Cu−O (1.54 Å) are almost unchanged after the introduction of Na (Fig. S15). Noteworthy, the intensity of the peaks belonging to the Ni−O (1.56 Å) bond in Na-HE LDH is weakened (Fig. 3d), which implies that there is a decrease in O coordinated to Ni, indicating the presence of a large number of Vo in the material, which is in agreement with the XPS and EPR results. HE LDH and Na-HE LDH were fitted to determine the Ni−O coordination number. The Ni K-edge EXAFS fitting curves (Fig. 3e and Table S3) show a six-coordination structure of Ni−O in HE LDH,

whereas the coordination number of the Ni−O bond in Na-HE LDH is 5, which suggests that the low-valence Na leads to the appearance of $V_O$ (Fig. 3f), which may be due to charge conservation.

To investigate the effect of Na on OER activity of HE LDH, we recorded the polarization curves of Na-HE LDH in $N_2$-saturated 1.0 M KOH solution. before of the test, we have purified the used electrolyte by a previous method to remove the effect of trace Fe[35,36]. The Fe content in the purified electrolyte was only 0.01 ppm (Fig. S16). Linear scanning voltammetry (LSV) curves showed that Na-HE LDH had the lowest overpotential (Fig. 4a). The overpotential of Na-HE LDH at a current density of 10 mA cm$^{-2}$ was only 176 mV, which was 154 and 145 mV lower than that of HE LDH (330 mV) and commercial $IrO_2$ (321 mV). We also confirmed that the increase in catalyst activity is attributed to Na doping rather than high entropy structure by Na doping in conventional binary LDH, as detailed in Supplementary Note 2 and Figs. S17, 18. Moreover, compared with HE LDH (54.8 mV dec$^{-1}$) and commercial $IrO_2$ (64.4 mV dec$^{-1}$), the Tafel slope (Fig. 4b) of Na-HE LDH decreased dramatically to 29.6 mV dec$^{-1}$, which attributed to the fact that the introduction of Na alters the OER mechanism of HE LDH by adjusting the OER rate-determining step and accelerating the reactant deprotonation[19,37]. In view of the low overpotential and Tafel slope, the prepared Na-HE LDH exhibits good OER activity than other recently reported advanced high-entropy electrocatalysts (Table S4). EIS results (Fig. S19) show that Na-HE LDH (2.8 Ω) has a much smaller charge transfer resistance compared to HE LDH (8.5 Ω), this indicates a accelerated charge transfer process at the solid-liquid interface of Na-HE LDH, further enhancing the OER kinetics[18]. We give LSV curves normalized by ECSA (Fig. S20 and Fig. S21) to exclude the effect of electrochemically active surface area (ECSA) on the intrinsic activity of the catalyst (ECSA = $C_{dl}/C_s$, see experimental section for details of the calculations)[7]. Without considering the contribution of the number of multiple active sites of Na-HE LDH to the

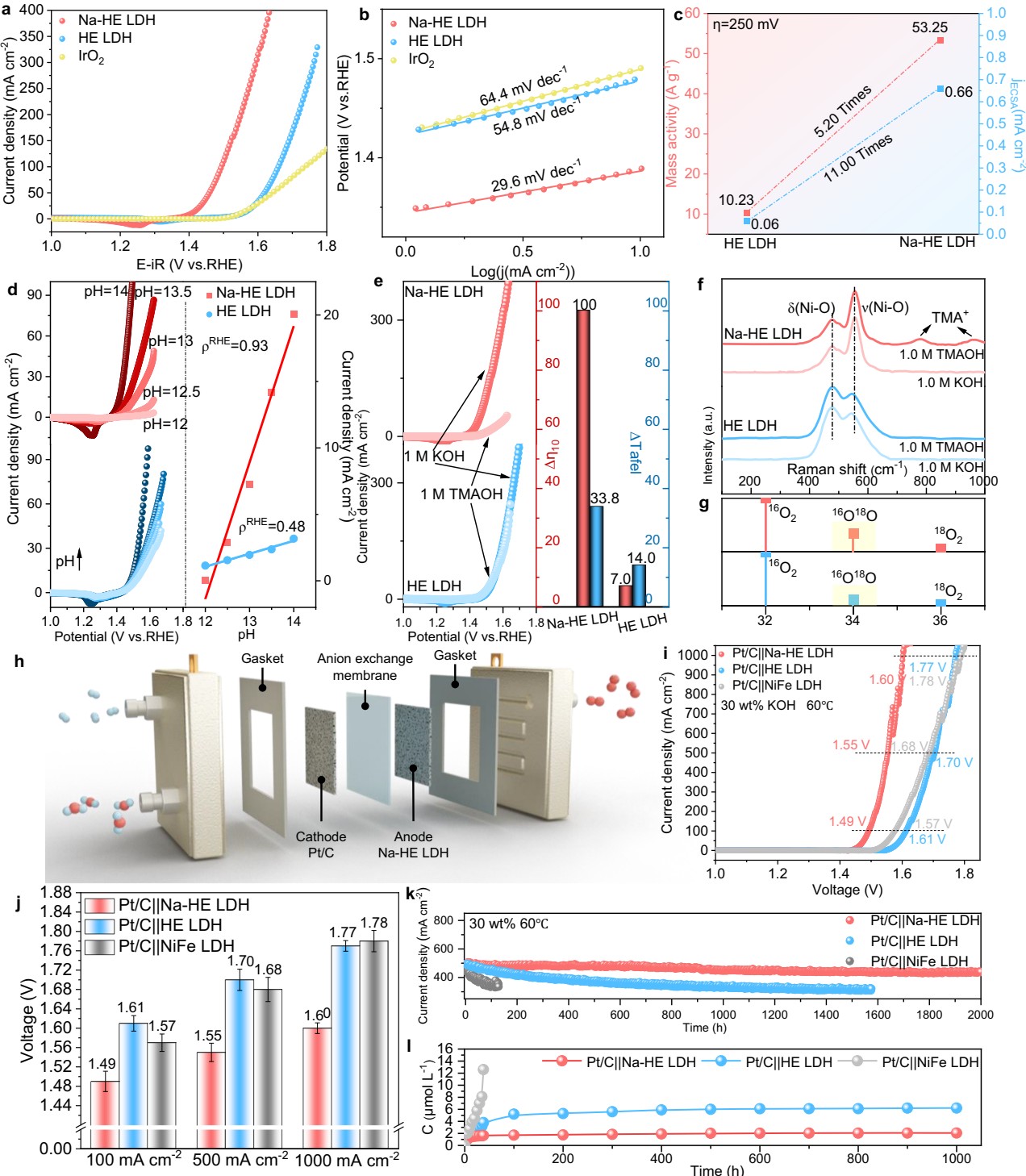

**Fig. 4 | Investigation of the OER performance and mechanism of Na-HE LDH.**
**a** LSV curves with 95% iR correction, the solution resistance is 1.8 Ω and the loading of catalysts is -1 mg cm⁻². **b** Tafel plots. **c** Mass activity and ECSA-normalized current density of Na-HE LDH and HE LDH at $\eta = 250$ mV. **d** left is the LSV curves with 95% iR correction in KOH solution with pH = 12–14, the solution resistance is 1.8 Ω and the loading of catalysts is -1 mg cm⁻². right is the proton reaction orders ($\rho^{RHE} = \partial \log j / \partial pH$). **e** left is the LSV curves with 95% iR correction of Na-HE LDH and HE LDH in 1.0 M KOH and 1.0 M TMAOH, the solution resistance is 1.8 Ω and the loading of catalysts is -1 mg cm⁻². right is the shift of overpotential ($\Delta\eta10$) and shift of Tafel slopes values ($\Delta$Tafel) from 1.0 KOH to 1.0 TMAOH. **f** Raman spectra of Na-HE LDH and HE LDH after

running in 1.0 M TMAOH and 1.0 M KOH (1.45 V vs RHE for 1 h). **g** Mass spectrometric results of ¹⁸O isotope labeling experiments. **h** Structure illustration of the electrolyzer cell. **i** LSV curves of Pt/C || Na-HE LDH, Pt/C || HE LDH and Pt/C || NiFe LDH in 30 wt% KOH at 60 °C, no iR correction. **j** Comparison of voltage values for Pt/C || Na-HE LDH, Pt/C || HE LDH and Pt/C || NiFe LDH at different current densities (error bars are from five repeated experiments). **k** i-t measurement for Pt/C || Na-HE LDH, Pt/C || HE LDH and Pt/C ||NiFe LDH at 1.56 V vs. RHE, 1.71 V vs. RHE and 1.69 V vs. RHE. l, Variation in the molar concentration of Fe in the electrolyte during the stability tests of Pt/C || Na-HE LDH, Pt/C || HE LDH and Pt/C ||NiFe LDH. Source data are provided as a Source Data file.

OER activity, Na-HE LDH still possesses much higher OER activity than the pristine HE LDH, suggesting that the introduction of Na improves the intrinsic activity of HE LDH. At $\eta = 250$ mV, the specific activity of Na-HE LDH was 0.66 mA cm$^{-2}$, which was 11.00 times higher than that of pristine HE LDH (0.06 mA cm$^{-2}$). In addition, the mass activity of Na-HE LDH (53.25 A g$^{-1}$) was 5.20 times higher than that of HE LDH (10.23 A g$^{-1}$) at $\eta = 250$ mV (Fig. 4c and Table S5). Similarly, the turnover frequency (TOF, Fig. S22) of Na-HE LDH at $\eta = 250$ mV is 0.052 s$^{-1}$, which is 14.4 times that of HE LDH (0.0036 s$^{-1}$). The above results indicate that the intrinsic activity of HE LDH can be enhanced by the introduction of the low-valent metal Na. The wettability of the catalyst is also crucial as large bubbles are usually generated quickly on the catalyst surface at high current density to block the active sites and hinder the mass transfer. As shown in Fig. S23, the droplet contacts angle of Na-HE LDH is 15.59°, which is obviously smaller than that of HE LDH (139.53°). This indicates that Na-HE LDH is more hydrophilic, which is conducive to mass transfer and elimination of air bubbles, and ensures stable operation at high current density[38,39]. The improved wetting properties of Na-HE LDH may be attributed to its rough nanosheet morphology, which promotes water adsorption. This was confirmed by the calculated water adsorption energies (Fig. S24), which were lower on Na-HE LDH (−0.836 eV) than on pristine HE LDH (−0.177 eV), as well as thermodynamically better hydrophilicity. Moreover, long-term Faraday efficiency (FE) measurements of Na-HE LDH were performed, the measured oxygen volume agrees well with the theoretical values calculated by Faraday's law of electrolysis, approaching 100% FE (Fig. S25)[40]. The above results indicate that the low-valent Na and in-situ generated Vo can increase the intrinsic activity of HE LDH. It should be noted that not only the activity, but also the stability is the key to determine whether the catalyst can be applied industrially. Excitingly, the synthesized Na-HE LDH has a robust OER stability. The current density ( ~ 500 mAcm$^{-2}$) decayed by only 7% after 1000 h of continuous operation at a constant potential of 1.65 V (Fig. S26), which may be attributed to the entropic stabilization effect and hysteresis-diffusion effect of high-entropy materials, as conventional binary LDHs deactivate rapidly at the early stage of the stability test, as detailed in Supplementary Note 3 and Fig. S27. The long-term stability of Na-HE LDH was also better than most LDH and oxyhydroxide OER catalysts (Table S6). Moreover, the OER performance of Na-HE LDH in 1.0 M NaOH electrolyte was also explored, see Supplementary note 4 and Fig. S28.

Considering the low Tafel slope of Na-HE LDH, we gained insight into the OER mechanism of Na-HE LDH. While conventional AEM contains four proton-synergistic electron transfer processes, LOM contains a non-synergistic proton-electron transfer step (deprotonation of hydroxyl groups without electron transfer), leading to pH dependence of the reversible hydrogen electrode (RHE)[41]. So, the dependence of OER activity on pH was investigated, and as shown in Fig. 4d, the current density of Na-HE LDH increased dramatically with pH increasing (from 12.5 to 14.0), suggesting a non-synergistic proton-electron transfer pathway. In contrast, the change in current density of HE LDH with pH increasing (from 12.5 to 14.0) is negligible. The pH dependence of the catalyst is quantified by the proton reaction level ($\rho^{RHE}$, $\rho^{RHE} = (\partial \log j / \partial pH)$), where j is current density at 1.45 V vs. RHE. The closer the $\rho$-value is to 1, the stronger the pH-dependent properties of the catalyst. Compared with the $\rho^{RHE}$ value of HE LDH (0.48), the $\rho^{RHE}$ value of Na-HE LDH (0.93) is closer to 1, indicating that Na-HE LDH exhibits a stronger pH-dependence during OER, which means that it tends to follow the LOM while HE LDH is more inclined to follow AEM[7]. Unlike conventional AEM, LOM generates negatively charged peroxide ($O_2^{2-}$) intermediate species during OER. To validate the LOM mechanism, tetramethylammonium cations (TMA$^+$) were introduced as chemical probes for recognizing $O^{2-}$ intermediates because they have strong electrostatic interactions and impede OER kinetics[42]. As shown in Fig. 4e and Fig. S29, the OER activity of Na-HE LDH was

reduced in 1.0 M TMAOH electrolyte ($\Delta\eta 10 = 100$ mV, $\Delta$Tafel=33.8 mV dec$^{-1}$). This reflects that the OER process of Na-HE LDH is severely hindered by TMA$^+$, verifying that the OER reaction of Na-HE LDH proceeds on the basis of LOM. In contrast, HE LDH exhibits negligible minor changes in both OER activity and kinetics ($\Delta\eta 10 = 7$ mV, $\Delta$Tafel=14.0 mV dec$^{-1}$), which suggests a AEM process for HE LDH. The presentation of $O_2^{2-}$ were further verified by Raman spectroscopy (Fig. 4f). HE LDH and Na-HE LDH were labeled in 1.0 M KOH and 1.0 M TMAOH electrolytes at 1.45 V vs. RHE for 1 h, washed with deionized water and used for Raman testing. Where the two major Raman peaks located at 400 ~ 600 cm$^{-1}$ belong to the $E_g$ bending vibrational mode ($\delta$(Ni−O)) and $A_{1g}$ stretching vibrational mode ($v$(Ni−O)) in the Ni−O band, respectively. Among them, typical characteristic peaks attributed to TMA$^+$ are observed at 766.1 cm$^{-1}$ and 959.7 cm$^{-1}$ for Na-HE LDH, but there is no TMA$^+$ peak for HE LDH. The above analysis proves that the OER mechanism of HE LDH is transformed from AEM to LOM after Na doping. To directly reveal the role of lattice oxygen in the OER process, $^{18}O$ isotope labeling experiments were carried out on Na-HE LDH and HE LDH, as described in the details of the Experiments section. Measurement of oxygen products (Fig. 4g) by gas chromatography-mass spectrometry (GC−MS) (signal intensity was normalized by $^{16}O_2$) revealed that Na-HE LDH has a high content of $^{16}O^{18}O$ products, whereas HE LDH has a negligible amount of $^{16}O^{18}O$ products, indicating that Na-HE LDH considerable amount of lattice oxygen in HE LDH is involved in the OER reaction. This implies that Na-HE LDH is more inclined to follow LOM during OER, while HE LDH is more inclined to AEM, confirming that the enhancement of the intrinsic activity of HE LDH is due to the alteration of the OER mechanism caused by the doped of Na.

Given the high activity of Na-HE LDH, we used a customized anion-exchange membrane electrolysis cell (AEMWE, Fig. 4h) to evaluate the value of Na-HE LDH for practical applications at 30 wt% KOH and 60°C. Pt/C was used as the cathode and Na-HE LDH as the anode to form an electrode pair (Pt/C||Na-HE LDH), and Pt/C||HE LDH and Pt/C||NiFe LDH were used for comparison. As shown in Fig. 4i, the AEM aqueous electrolyzer based on Pt/C||Na-HE LDH requires only cell voltages of 1.49 V, 1.55 and 1.60 V to achieve high current densities of 100 mA cm$^{-2}$, 500 mA cm$^{-2}$ and 1000 mA cm$^{-2}$. This is better than the use of Pt/C||HE LDH and Pt/C||NiFe LDH electrolyzers (Fig. 4j). More importantly, the Pt/C||Na-HE LDH electrolyzer also exhibits good long-term stability when tested continuously for 2000 h at ~500 mA cm$^{-2}$ (Fig. 4k). Given the high electrical conductivity and alkali resistance of nickel foam (NF), we coated the catalyst on NF to assemble the AEMWE, and the Pt/C||Na-HE LDH electrode pair still maintains high activity and stability, as detailed in Supplementary Note 5 and Fig. S30. This confirms the potential and advantages of Pt/C||Na-HE LDH for large-scale industrial hydrogen production. In the case of industrial alkaline water electrolysis, the use of Na-HE LDH electrocatalysts is advantageous because it uses a large number of highly abundant and low-cost elements (Table S7). Considering the estimated manufacturing cost of the catalyst, which totaled 0.145 \$ mg$^{-1}$, lower than current commercial catalysts such as IrO$_2$ (0.702 \$ mg$^{-1}$), Na-HE LDH emerged as a promising alternative to precious metal catalyst

Previous studies have shown that Fe-based LDH have poor durability due to attenuation of catalyst activity due to distortion of the FeO$_6$ lattice at high oxidation potentials leading to breakage of the Fe +9*O bond and leaching of Fe in the lattice[36,43]. In view of the robust durability of Na-HE LDH in anion-exchange membrane electrolyzes, investigating the mechanisms by which high-entropy materials and Na introduction affect the enhancement of the stability of Fe-based LDH. We monitored the changes in Fe concentration in the electrolyte over a long period of stability testing of individual samples and analyzed them using ICP-MS (Fig. 4l). The molar content of Fe in the electrolyte continued to rise for the tests as Fe continued to dissolve out of the

NiFe LDH over time. The Fe content in the electrolyte was as high as 12.6 μmol L$^{-1}$ after only 38 h of operation, the current density of NiFe LDH decreases dramatically and the structure is degraded, indicating that NiFe LDH undergoes severe Fe dissolution at high OER potentials, as shown in Fig. 4l. Compared to NiFe LDH, the amount of Fe dissolved in HE LDH and Na-HE LDH was reduced. The concentration of Fe in the electrolytes of Na-HE LDH and HE LDH increased slowly at the early stages of stability testing, and the molar concentrations reached 2.9 and 1.6 μmol L$^{-1}$ at 30 h, respectively. Both subsequently remained at comparable levels, indicating that Fe leaching ceased after this period. This explains why Na-HE LDH and HE LDH can maintain good stability under industrial conditions, with the high entropy effect enhancing the metal-oxygen bonding and inhibiting the Fe leaching[21,44]. In addition, the introduction of Na further anchors Fe, enhancing the structural and catalytic stability of the structures. We performed XRD, HRTEM, mapping, XPS and EPR tests on Na-HE LDH after stability test. The results show that Na-HE LDH still maintains a crystalline hexagonal hydrotalcite structure (Fig. S31) and still has a nanosheet morphology (Fig. S32), but do not exclude the presence of a surface metallic amorphous layer. HRTEM (Fig. S33) observed an amorphous layer of about 20 nm, and the mapping (Fig. S34) results showed that the Mn, Fe, Co, Ni, Cu and Na elements were still uniformly distributed without obvious aggregation and segregation. In addition, the XPS results (Fig. S35) confirm that elements such as Mn, Fe, Co, Ni, Cu and Na are still present. while the EPR results (Fig. S36) confirm the retention of the Vo. More importantly, compared with the conventional NiFe LDH (28.9%), the leaching rate of Fe ions in HE LDH (3.2%) and Na-HE LDH (1.1%) was greatly reduced and Na-HE LDH had the lowest leaching rate of Fe ions (Fig. S37), which further confirmed the robust stability of Na-HE LDH.

## Monitoring the dynamic evolution of metal sites during OER process

Electronic interactions between polymetallic atoms play a crucial role in the construction of highly active centers to accelerate reaction kinetics, especially in terms of a fundamental understanding of the true catalytic sites and structural evolution in catalytic systems. To understand the catalyst surface evolution, we monitored the catalyst dynamics using electrochemical in situ Raman spectroscopy. The initial HE LDH and Na-HE LDH were identified with two characteristic peaks located at 466.4 and 546.3 cm$^{-1}$, respectively, and these two characteristic Raman signals can be indexed as $E_g$ bending vibration ($\delta$(Ni−O)) and $A_{1g}$ stretching vibration ($\nu$(Ni−O)) modes in Ni−O (Fig. 5a–d)[45]. With the increase of the applied voltage, the two characteristic peaks are blue-shifted and transformed from Ni$^{II}$-O to Ni$^{III}$-O[46]. Generally, Ni$^{III}$-O contains β and γ phases[19]. The higher $\delta$(Ni−O) (i.e., higher $I_{\delta(Ni−O)}/I_{\nu(Ni−O)}$) in $\gamma$(Ni$^{III}$-O) is attributed to its higher disorder and higher valence of Ni[47,48]. It is worth noting that the $I_{\delta(Ni−O)}/I_{\nu(Ni−O)}$ of HE LDH and Na-HE LDH are different. As can be seen from Fig. 5e, the $I_{\delta(Ni−O)}/I_{\nu(Ni−O)}$ values of Na-HE LDH increase with the increase of the applied bias voltage. It indicates that with the increase of voltage, the $\gamma$(Ni$^{III}$-O) structure is more easily formed when the surface of Na-HE LDH is oxidized, and usually the $\gamma$(Ni$^{III}$-O) structure exists with highly oxidized Ni, and the highly oxidized state of Ni is favorable for OER[49,50]. As the applied bias continues to increase, the two distinct characteristic peaks of HE LDH and Na-HE LDH are gradually shifted and broadened until they merge into one large peak, a process that represents the elevated oxidation state of Ni and the generation of surface amorphous layers (Fig. 5a–d). It is noteworthy that the potential for the generation of an amorphous layer on the surface of Na-HE LDH (1.55 V) is higher than that of HE LDH (1.35 V), suggesting that the introduction of Na further enhances the binding of the metal-oxygen bonding in the HE LDH and improves the stability of HE LDH.

To further reveal the structural changes and electron transfer of Na-HE LDH and HE LDH during the OER process, the in situ XAS spectra

of Na-HE LDH and HE LDH at different applied potentials (1.1 ~ 1.7 V vs. RHE) were recorded (Fig. 5f and i). The intensity of the white line peak was measured as a function of the applied potential, quantifying the number of empty orbital states and thus the oxidation state of each element in Na-HE LDH and HE LDH. The in situ XANES spectra show that the valence states of Ni in Na-HE LDH and HE LDH increase from the open-circuit potential (OCV) to 1.7 V (Fig. 5g and j) whereas the valence states of Mn, Fe, Co, Ni and Cu increased slightly (Figs. S38, S39). We further give the values of valence changes with increasing voltage for all metallic elements (Figs. S40, S41), where the valence of all the metal elements increases with increasing oxidation potential, suggesting that the HE LDH and Na-HE LDH are reconstruction. It is noted that among all the elements of HE LDH, the Ni has the largest change in valence state (0.45, from +2.71 to +3.16), and the Ni also has the largest change in valence in Na-HE LDH (0.48, from +2.77 to +3.25). This indicates that Ni involved in the OER process through electron transfer and is evaluated as the active site. It is important to note here that although the LOM mechanism is O as the active site, there is also a part of the AEM mechanism in the whole OER, and Ni is the active site in the AEM mechanism, which leads to the drastic change of Ni valence. It is noteworthy that the valence state of Ni is higher in Na-HE LDH compared to HE LDH, which is more favorable to the OER process, consistent with the in-situ Raman results. Furthermore, according to the in-situ EXAFS spectra of Na-HE LDH and HE LDH, the bond lengths of the M−O bonds in Na-HE LDH become slightly shorter with the increase of the applied potential (Fig. S42), especially the Fe−O bonds (Fig. 5h). It is well known that Fe usually dissolves by Fe−O bond breaking at high oxidation potentials of OER, resulting in poor stability of Fe-based catalysts. The shortening of Fe−O bonds confirms that Na enhances the bonding of M−O bonds in HE LDH, which is very favorable for the stability of the structure. In contrast, the M−O bonds in HE LDH became longer with increasing voltage (Fig. 5k and Fig. S43), which was detrimental to the structural stability of the catalyst.

## Theoretical insights into OER mechanism

To gain insight into the intrinsic reasons for the OER pathway of the low-valent Na-converted HE LDH, DFT calculations were performed. Based on the results of the above section in situ experiments, we selected Na-HEOOH instead of Na-HE LDH, HEOOH as a comparison (Fig. S44). We calculated the total density of states (TDOS) (Fig. S45) and found that the DOS near $E_f$ becomes more continuous and intensified after Na doping, suggesting a potential enhancement in electronic conductivity, which may qualitatively correlate with the improved charge transfer behavior observed in EIS measurements. Considering that the distance between the O 2$p$ band center and the Fermi energy level ($E_f$) is an important parameter for identifying the lattice oxygen activity, in order to reveal the activity of lattice oxygen in Na-HEOOH and HEOOH, we calculated the density of states of the O 2$p$ and M 3$d$ orbitals (Fig. 6a). We estimate the O 2$p$ band centers ($\varepsilon$O-2$p$) of Na-HEOOH (− 2.5997 eV) and HEOOH (− 2.6744 eV), and find that the $\varepsilon$O-2$p$ of HEOOH is closer to the $E_f$ after the introduction of Na, which may promote the release of lattice oxygen from the lattice in favor of the LOM. Moreover, we observed a notably enhanced overlap between the O 2$p$ and M 3$d$ states in Na-HEOOH compared to HEOOH, reflecting a stronger orbital hybridization. This increased overlap is indicative of enhanced M−O covalency, which is beneficial for promoting electron delocalization and is generally correlated with enhanced lattice oxygen activity[13,51]. Fig. 6b further shows that the O−O coupling in the LOM caused by the introduction of Na can potentially reduce the unoccupied $O_{NB}$ portion around the $E_f$, which may contribute to improved electronic stabilization[52]. According to the molecular orbital theory, M−O bonds and M−O* antibonding bands are formed between M and O in MOOH, and they have oxygen and metal features, respectively. The strong d-d Coulomb interaction in the M−O* antibonding orbital produces a Mott-Hubbard split, forming an empty

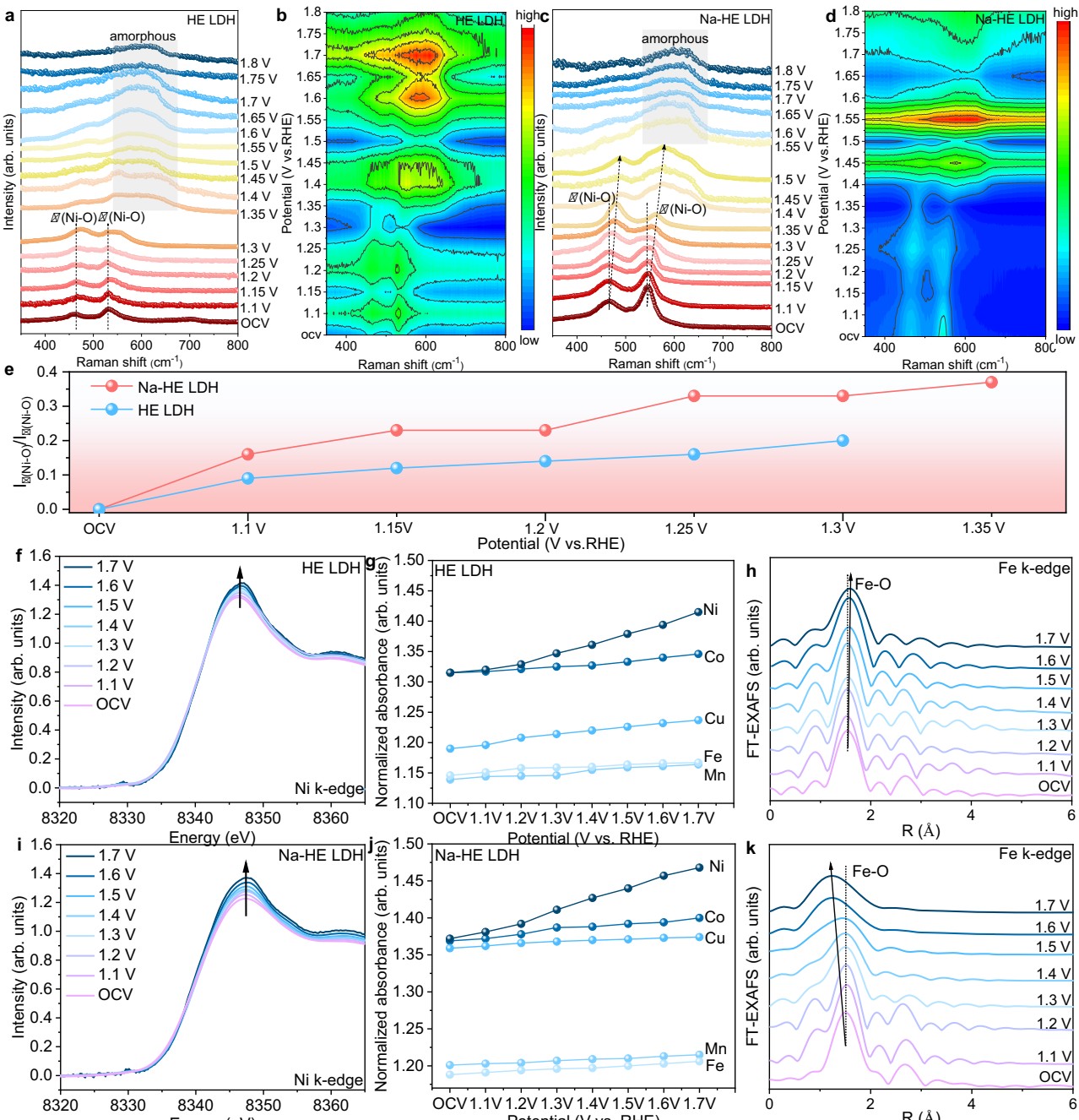

**Fig. 5 | Structural evolution of Na-HE LDH during the OER process.** In-situ Raman spectra and the corresponding contour plots of the in-situ Raman spectra for **a**, **b** HE LDH and **c**, **d** Na-HE LDH. **e** δ(Ni–O) to ν(Ni–O) ratios in electrochemical in-situ Raman spectra of Na-HE LDH and HE LDH related to the operated potentials (Normalized by the respective open-circuit voltage). **f** In-situ Ni k-edge XANES spectra of HE LDH. **g** Changes in the normalized absorbance of 3d transitional elements in the in-situ k-edge XANES spectra of HE LDH. **h** Fourier transforms of operando Ni k-edge EXAFS spectra of HE LDH, R is the radial distance. **i** In-situ Ni k-edge XANES spectra of Na-HE LDH. **j** Changes in the normalized absorbance of 3 d transitional elements in the in-situ k-edge XANES spectra of Na-HE LDH. **k** Fourier transforms of operando Ni k-edge EXAFS spectra of Na-HE LDH, R is the radial distance. Source data are provided as a Source Data file.

upper Hubbard band (UHB) and an electron-filled lower Hubbard band (LHB)[17,18]. The energy difference (ΔU) between the UHB and LHB centers has been proposed as a possible descriptor for evaluating the lattice oxygen activity[53]. Compared to HEOOH, the M–O in Na-HEOOH all have larger ΔU (Fig. 6b and Table S8), and the larger ΔU leads to a downward shift of the LHB, enhancing the covalency of the M–O bonds to ensure that electrons are removed from the $O_{NB}$ rather than from the LHB[54]. With the elimination of the unoccupied oxygen state in Na-HEOOH, the OER mechanism switches to LOM. In addition, as can be

seen from Fig. 6c and Table S9-S10 the LHB center value (-3.015 eV) of the Ni–O bond is the most negative of all M–O LHB centers in Na-HEOOH (Mn–O, −2.989 eV; Fe–O, −2.997 eV; Co–O, −2.969 eV; Cu–O, −2.977 eV) indicating that the Ni–O bond is the is the weakest, so the lattice oxygen near Ni is more likely to be oxidized as an active center and escape as $O_2$[18]. Fig. 6d, e shows the projected density of states (PDOS) plots of the Ni 3d orbitals and O 2p orbitals bonded with Ni in Na-HE LDH and HE LDH. σ-bonding arises from the overlap of the Ni $3d_z^2$ and O $2p_z$ orbitals. Unlike HEOOH, Na-HEOOH has more σ bonding

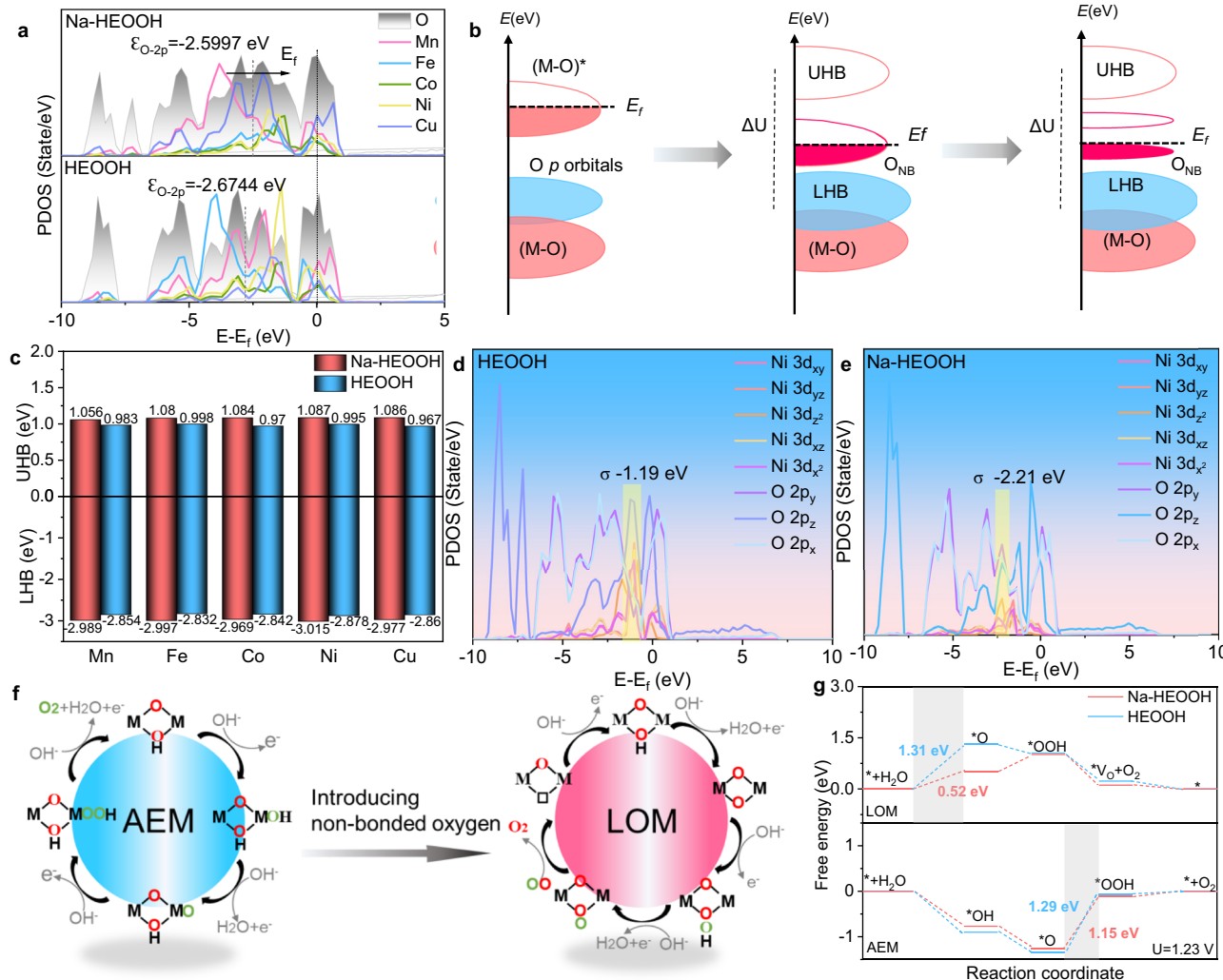

**Fig. 6 | DFT calculation. a** Projected density of states. **b** Switching of the OER mechanism to the LOM with the elimination of unoccupied oxygen states in Na-HEOOH. **c** The UHB and LHB center positions of Na-HEOOH and HEOOH. Projected density of states of Ni sites and O sites after oxygen adsorption of **d** Na-HEOOH and **e** HEOOH. **f** AEM and LOM on HEOOH (red oxygen is lattice oxygen, green oxygen is adsorbed oxygen and □ is oxygen vacancy). **g** Calculated Gibbs free energy (ΔG) of OER based on AEM and LOM of Na-HEOOH and HEOOH. Source data are provided as a Source Data file.

electron filling, which implies a weaker coupling between Ni and O. This further suggests that O around Ni is more likely to break bonds to be involved in the OER process.

Usually, the conventional AEM goes through four steps containing *OH, *O, *OOH three intermediates, while the LOM goes through five steps containing (*O, *OOH, *OO and Vo four different intermediates, as shown in Fig. 6f. Firstly, the adsorption free energies of the OER intermediates on the 3d transitional elements sites in Na-HEOOH and HEOOH were calculated based on AEM (Fig. S46, S47 and Table S11). The results suggest that the Ni sites exhibited the lowest energy barriers (1.41 eV and 1.76 eV) in the AEM pathway. Gibbs adsorption free energy diagrams for Na-HEOOH and HEOOH are given based on the AEM (Ni as the active site) and LOM (O in the near of Ni as the active site) pathways (Fig. 6g, Figs. S48, S49 and Table S12). Deprotonation of *OH in the AEM pathway of the Na-HEOOH and HEOOH is rate-determining steps with potentials of 0.7 eV and 1.29 eV, respectively. The first electrochemical deprotonation step in the LOM pathway is the rate-determining step for Na-HEOOH and HEOOH, with energy potentials of 0.52 eV and 1.31 eV, respectively. From a thermodynamic point of view, the Na-HEOOH tends to follow the LOM while HEOOH tends to follow the AEM, which is qualitatively consistent with trends observed in experimental studies.

To elucidate the effect of high entropy on the activity and stability of OER from the perspective of theoretical study, we constructed a high-entropy model with disordered metal arrangement and a non-high-entropy model with ordered metal arrangement, as shown in Fig. S50. We calculated the formation energies of the two structures (Fig. S51). The results suggest that the formation energy of the high-entropy structure (−1.59 eV) is lower compared to that of the non-high-entropy structure (−0.41 eV), indicating that the high-entropy structure has higher thermodynamic stability. In addition, we also evaluated the OER energy barrier on the AEM pathway (Fig. S52) and found that the energy barrier of the high-entropy structure (1.29 eV) is lower than that of the non-high-entropy structure (1.50 eV), which suggests that its intrinsic OER activity is enhanced. These theoretical results suggest that the polymetallic synergistic effect inherent in the high-entropy structure indeed contributes to the enhancement of stability and activity.

## Discussion
In conclusion, we present a sodium-doped high-entropy layered double hydroxide catalyst with in situ generated oxygen vacancies, achieving enhanced OER activity compared to pristine HE LDH while demonstrating good long-term durability under industrial conditions.

Advanced in situ spectroscopic techniques and DFT + U calculations reveal that the introduction of Na ions and oxygen vacancies facilitates the formation of oxygen non-bonding bands. This innovation enables redox reactions to occur within these non-bonding bands rather than the conventional M−O bands, overcoming the limitations of the traditional adsorbate evolution mechanism. Moreover, this approach greatly enhances the structural and catalytic stability of the material. This study provides a valuable framework for the design of efficient and stable lattice oxygen catalysts, paving the way for their application in large-scale water electrolysis processes.

## Methods

### Materials
$Ni(NO_3)_2 \cdot 6H_2O$ (99%), $Fe(NO_3)_3 \cdot 9H_2O$ (99%), $Co(NO_3)_2 \cdot 6H_2O$ (99%), $Mn(NO_3)_2 \cdot 4H_2O$ (98%), $Cu(NO_3)_2 \cdot 3H_2O$ (99%), NaCl (99.99%), $NH_4F$ (99.9%), urea (99%), $IrO_2$ (99.9%) and KOH (95%) were obtained from Macklin. 1.0 M KOH electrolyte is prepared as needed.

### Preparation of HE LDH and Na-HE LDH
The HE LDH was synthesized using a hydrothermal approach. Initially, 0.45 mmol each of $Ni(NO_3)_2 \cdot 6H_2O$, $Fe(NO_3)_3 \cdot 9H_2O$, $Co(NO_3)_2 \cdot 6H_2O$, $Mn(NO_3)_2 \cdot 4H_2O$, and $Cu(NO_3)_2 \cdot 3H_2O$ were dissolved in 35 mL of deionized water. To this solution, 4 mmol of $NH_4F$ and 10 mmol of urea were added, which serve to control the nucleation rate and regulate the nanosheet morphology. After ensuring complete dissolution, the mixture was transferred into an autoclave, along with a pre-treated piece of carbon cloth (CC), and heated at 120 °C for 6 h. $Na_{0.045}$-HE LDH (labeled as Na-HE LDH in the text) was obtained by adding 0.045 mmol NaCl to the precursor, otherwise consistent with the above steps. In addition, $Na_{0.025}$-HE LDH and $Na_{0.065}$-HE LDH were obtained by adding 0.025 mmol NaCl and 0.065 mmol NaCl to the precursor. The loading of all the catalysts is ~1 mg cm$^{-2}$.

### Structural characterization
X-ray diffraction (XRD) patterns were recorded using a Bruker D8 Advance diffractometer (Bruker, Germany) with Cu-Kα radiation (40 kV, 40 mA, λ = 0.154178 nm) at a scan rate of 6° min$^{-1}$. X-ray photoelectron spectroscopy (XPS) was conducted with a Kratos XSAM 800 spectrophotometer. Sample morphologies and compositions were analyzed using a Tecnai G20 U-Twin transmission electron microscope (TEM) equipped with an energy-dispersive X-ray detector (EDX) at 200 kV. Scanning electron microscopy (SEM) images were obtained using a SIGMA microscope (ZEISS). Raman spectra were acquired with a HORIBA Raman microscope using a 532 nm laser. Inductively coupled plasma mass spectrometry (ICP-MS) was performed on a Thermo IRIS Intrepid II XSP spectrometer. X-ray absorption fine structure (XAFS) measurements were carried out at the 02B02 beamline of the Shanghai Synchrotron Radiation Facility (SSRF), with the data processed in Athena for background correction and edge calibration. Atomic force microscopy (AFM) imaging was conducted on a 5500 AFM/STM system.

### Electrochemical measurements
The electrochemical characterization was performed using a CHI 660E electrochemical workstation, operating at ambient temperature with a three-electrode setup. The working electrode was the Na-HE LDH, HE LDH and $IrO_2$ etc. (electrode area are $1 \times 1$ cm$^2$, catalyst loading ~ 1 mg cm$^{-2}$), while a carbon rod served as the counter electrode and Hg/HgO as the reference electrode. We continued to calibrate the Hg/HgO electrode before testing, using Pt wire as the working/counter electrode. First, high-purity hydrogen gas was passed into a 1.0 M KOH (pH = 13.9 ± 0.1) electrolyte for 30 min, followed by scanning using cyclic voltammetry (CV) with a scanning range of -1.2 ~ 0 V vs. Hg/HgO and a scanning rate of 1 mV s$^{-1}$. Samples were activated and stabilized by multiple cyclic voltammetry (CV) curves across a potential range of

0 ~ 0.8 V vs. Hg/HgO at 100 mV s$^{-1}$ and 20 mV s$^{-1}$ before Linear sweep voltammetry (LSV) curves measure. LSV curve was performed in a 1.0 M KOH solution, scanning at 2 mV s$^{-1}$ across a potential range of 0.8 ~ 0 V vs. Hg/HgO with 95% iR compensation, the solution resistance is 1.8 Ω. The potential was converted to the reversible hydrogen electrode (RHE) scale using the Nernst equation: $E_{RHE} = E_{Hg/HgO} + 0.059 \times pH + 0.095$ (1). Electrochemical impedance spectroscopy (EIS) was conducted between 0.01 ~ 100 kHz at 0.6 V vs. Hg/HgO. Cyclic voltammetry (CV) was used to measure double-layer capacitance ($C_{dl}$) and estimate the electrochemically active surface area (ECSA, ECSA=$C_{dl}/C_s$ (2)), with scan rates ranging from 5 to 100 mV s$^{-1}$ within a potential window of 0.14 ~ 0.24 V vs. Hg/HgO. A standard specific capacitance ($C_s$) of 40 μF cm$^{-2}$ was applied for these calculations.

Turnover frequency (TOF) calculations:

$$TOF = (j \times A)/(2 \times n \times F) \tag{1}$$

where j is the measured current density, A is geometric area of the electrode, F is the Faraday constant (96485 C mol$^{-1}$). n is the number of moles of Ni on the HE LDH and O on the Na-HE LDH.

### Isotope labeling experiments
Na-HE LDH and HE LDH were first cycled in a 1.0 M KOH solution containing $^{18}O$ over a potential range of 0 to 0.8 V vs Hg/HgO to incorporate the $^{18}O$ isotope. Following this, the samples were thoroughly rinsed with deionized water and dried. These $^{18}O$-labeled samples were then subjected to continuous testing at 1.45 V vs RHE for 1 h, during which the evolved gas was collected for differential electrochemical mass spectrometry (DEMS) analysis. To account for the natural abundance of $^{18}O$, the mass signal of $^{34}O_2$ was normalized against the total signal intensity of $^{36}O_2$ recorded in the same experiment.

### Assembly of anion exchange membrane water electrolysis
The anion exchange membrane water electrolyze system consists of four components: power supply, temperature control system, flow control system, and anion exchange membrane water solution. The electrolyser consists of two corrosion-resistant TA1 grade plates with SUS 316 end plates and a curved flow field machined into the bipolar plates. The anode and cathode electrodes are separated by an anion-exchange membrane to prevent gas crossings and are pressed against the sides of the electrolyze to form a membrane electrode assembly (MEA). The MEA was made of an anion exchange membrane (FAA-3-50, thick of 50 μm) with cathode (Pt/C, an area of 4 cm$^2$ and a loading of ~1 mg cm$^{-2}$) and anode (Na-HE LDH, an area of 4 cm$^2$ and a loading of ~1 mg cm$^{-2}$) catalysts loaded on a carbon cloth and located on both sides of the membrane. The temperature of the electrolytic cell was maintained at 60 °C, and 30 wt% KOH was used as the electrolyte. A peristaltic pump delivered the electrolyte into the electrode channel at a flow rate of 100 sccm.

### DFT calculations
All density functional theory (DFT) calculations were carried out using the Vienna Ab initio Simulation Package (VASP)[55,56]. The computations employed the projector augmented wave (PAW)[56,57] pseudopotential in conjunction with the Perdew-Burke-Ernzerhof (PBE)[58] generalized gradient approximation (GGA)[59] exchange-correlation functional. The energetics of metal oxides were determined using DFT with the Hubbard-U approach (DFT + U) to accurately describe the strongly localized d-electrons in 3 d metals. The Hubbard-U values applied were 3.9, 4.0, 3.3, 6.0 and 3.87 eV for Mn, Fe, Co, Ni, and Cu, respectively. The plane wave basis set had a cutoff energy of 450 eV, and a $3 \times 3 \times 1$ Monkhorst-Pack mesh was used for K-point sampling. All structures were spin-polarized, and full atomic relaxation was performed with an energy convergence threshold of 10$^{-5}$ eV per atom.

## Data availability

The source data generated in this study are provided in the Source Data file. The computationally optimized atomic model is publicly available at the repository (ref. 60). Source data are provided with this paper.

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

## Acknowledgments

The authors acknowledge the Analytical and Testing Center, Shandong University of Technology. This study was supported by the National Natural Science Foundation of China (22274083, H.L.C.), the Shandong Provincial Natural Science Foundation (ZR2023LZY005, H.L.C.), and the Exploration project of the State Key Laboratory of BioFibers and Eco-Textiles of Qingdao University (TSKT202101, H.L.C.).

## Author contributions

F.Q.W. designed the experiments, prepared the materials, performed most of the characterizations, conducted the theoretical calculations and drafted the manuscript. L.F. and M.W.Z. conducted material characterization. H.L.C. contributed to extensive revisions and has provided financial support. All the co-authors contributed to the discussion and commented on the manuscript.

## Competing interests

The authors declare no competing interests.
