## [Transparent Peer Review file · Nature Communications]

Engineering Oxygen Nonbonding States in High Entropy Hydroxides for Scalable Water Oxidation

Corresponding Author: Professor Hailin Cong

Version 0:

Reviewer comments:

Reviewer #1

(Remarks to the Author)

In this study, the authors reported a Na-doped HE LDH system and investigated the impact of Na on the formation of the oxygen non-bonding band in HE LDH towards enhanced OER activity. The authors conducted various characterizations and electrochemical studies. Significantly, the reported OER overpotential and long-term stability at high current density are interesting. However, addressing the identified issues will further strengthen the manuscript. So, after substantial revisions, the novelty and significance of this work are recommendable for publication in "Nature Communications" journal.

The significant concerns of this work are as follows.

1. Why did the authors choose a complex system of HE hydroxides to study the formation of the oxygen non-bonding band? This point should be stated clearly.
2. Carbon cloth is not typically recommended as a substrate for industrial-scale water electrolysis. Why did the authors use it in this study?
3. What do the authors mean by "industrial-scale electrolysis"? A 4 cm² area is generally not considered industrial scale. Could the authors use a larger electrolyzer to support their claim better? Additionally, how was the catalyst loading of 1 mg/cm² determined?
4. Have the authors conducted a cost analysis of their system? How does it compare to conventional systems of the same scale?
5. What is the thickness of the nanosheets based on SEM and TEM analysis? I recommend that the authors provide low-magnification SEM and TEM images to confirm the uniformity of the nanosheets on each carbon fiber in the CC support.
6. Why didn't the authors include oxygen TEM elemental mapping? The TEM elemental mapping of Na atoms is unclear in all the provided images. I recommend that the authors present more explicit images with more distinguishable colors.
7. The authors should provide more detailed information about the AEMWE assembly and its conditions.
8. How about the performance and stability of the Na-HE LDH in other electrolytes?
9. The authors recommend comparing their performance with other high-entropy systems of similar composition in the main manuscript, such as *Surfaces and Interfaces* 46 (2024) 104084, *Adv. Funct. Mater.* 2023, 33, 2301153, *J. Mater. Chem. A*, 2021, 9, 16841-16851, *Nat Commun* 14, 6019 (2023).
10. Did the authors conduct any control experiments on the Na doping by varying the Na content?
11. Interestingly, the oxidation states of Mn, Fe, Co, and Cu remain unchanged while Ni undergoes significant oxidation during OER. How do the authors explain it?
12. What is the influence of the amorphous layer on the long-term catalytic performance of Na-HE LDH?

Reviewer #2

(Remarks to the Author)

It is my pleasure to participate in the peer review of this manuscript for Nature Communications. The article provides a comprehensive analysis and mechanistic interpretation of a sodium-incorporated high-entropy Mn-Fe-Co-Ni-Cu hydroxide for alkaline water electrolysis. The material demonstrates exceptional performance with an overpotential of 176 mV at 10 mA cm⁻² and a remarkable durability of 2000 hours at a current density of 500 mA cm⁻². The study proposes that the oxygen evolution follows the lattice oxygen mechanism (LOM) with oxygen sites adjacent to Ni centers acting as active sites,

supported by thorough mechanistic investigations. While this work shows promise in addressing the stability challenges typically associated with LOM systems, as a reviewer, I would like to raise several critical questions and observations to ensure scientific rigor:

1. Has the impact of adding different sodium concentrations on performance been explored to obtain the optimal sodium concentration? The process of exploring the optimal concentration needs to be seen in the supporting materials.
2. The full spectrum of XPS analysis should be displayed in the material.
3. The reasons for adding higher concentrations of NH_4F and urea during the preparation process and their respective effects.
4. In Figure 4k, it should be indicated in the graph that the change in long-term test is at 1.7 V. The current annotation should be incorrect and difficult to understand. And this testing method for controlling voltage is somewhat inconsistent with the previous statement that the control current remains unchanged after 2000 hours of continuous operation at 500 mA cm^{-2} .
5. Can you conduct more in-depth research on the amorphous layer generated on the surface of the material, such as capturing its structure through transmission electron microscopy?
6. The author states that O near Ni acts as the active site, while when calculating the conversion frequency TOF, n is the number of moles of Ni on the electrode. Is this unreasonable?

Reviewer #3

(Remarks to the Author)

Wang et al. propose an ONB engineering strategy to address structural degradation in LOM-mediated reactive hydroxides. The synthesized ortho-hexagonal Na-HE LDH demonstrates promising electrocatalytic performance, achieving an overpotential of 176 mV at 10 mA cm^{-2} and maintaining stability for 2000 hours at 500 mA cm^{-2} in AEMWE applications. In addition, the contributions of LOW for Na-HE LDH have been quantified by TMAOH. While the authors executed comprehensive structural characterization protocols, the discussion section lacks atomic-scale mechanistic insights into how the introduced ONB features govern electrocatalytic. Overall, while the experimental characterization demonstrates technical competence, the work suffers from the superficial interpretation of characterization data and lacks hierarchical analysis bridging nano/microstructural features to macroscopic catalytic performance. I therefore recommend rejection in its current form.

1. The authors need to explain the underlying considerations of the selection of HE LDH, is the high-entropy configuration truly essential? Could the performance improvement be solely attributed to Na doping rather than the high-entropy architecture? The independent contributions of these two factors lack sufficient experimental validation.
2. The authors only detected the vibrational modes of Ni-O species in the Raman spectrum, can you explain that? The layer number reduction (5 \rightarrow 4) induced by Na incorporation in HE LDHs necessitates mechanistic interpretation.
3. The M-O covalency is strongly correlated to the LOM, while in this manuscript, the O K-edge has not been characterized and the variation of the M-O covalency has also not been discussed in deep. Moreover, the electronic configuration/chemical state differences between Na-HE LDH and HE LDH have not been reasonably explained.
4. Na-HE LDH shows a higher charge transfer rate and a better hydrophilic ability than He LDH, while the authors have not discussed these differences. There are many similar structural description deficiencies in the manuscript, the authors need to rigorously revise their manuscript.
5. Post-stability-test material characterization (e.g., structural integrity, surface composition) and quantitative metal leaching data should be supplemented to substantiate the stability conclusion.
6. The current DFT calculations appear oversimplified, as they fail to adequately capture the complex multi-metal synergistic effects inherent to high-entropy systems, which raises concerns similar to question #2.

Reviewer #4

(Remarks to the Author)

In this manuscript, Wang and coauthors synthesized sodium-doped high-entropy hydroxide (Na-HE LDH) as an oxygen evolution reaction (OER) catalyst in an alkaline environment, demonstrating good performance and stability in both RDE and WE tests. The reviewer would like to specifically address the XAS and electrochemical test results and analysis.

1. In Figure 3d-f, the authors claim that the coordination number of Ni is lower in Na-HE LDH compared to pure HE LDH based on FT-EXAFS data and fitting. However, the reviewer has several concerns regarding this analysis. First, the FT-EXAFS data for Na-HE LDH and HE LDH (Figure 3d) exhibit a noticeably different pseudo shoulder peak (slightly below 1 Å), which suggests an improper or inconsistent selection of the Fourier transform window and range during data processing. To ensure the validity of the analysis, the reviewer requests that the authors provide the raw EXAFS data and k-weighted spectra in the Supplementary Information and conduct a more rigorous analysis to rule out potential artifacts. Second, in EXAFS fitting, extracting coordination information requires careful consideration, as the coordination number and Debye-Waller factor (which represents the mean square displacement of atoms in the lattice) are correlated. This concern is particularly relevant here, as the authors report different Debye-Waller factors for Na-HE LDH and HE LDH (1.7 vs. 2.1),

suggesting that the fitting analysis may be problematic. The reviewer recommends analyzing the K-edge EXAFS of other metal species to determine whether the Debye-Waller factors of the two catalysts differ systematically. Additionally, the pseudo peak in the fitted curves appears significantly different, indicating that certain fitting parameters may have been set inconsistently. The reviewer urges the authors to carefully reassess their analysis to eliminate potential errors and ensure the reliability of their conclusions.

2. When extracting metal oxidation states from edge energy, an important assumption is that the metal has a similar coordination environment across samples. Therefore, metallic compounds are not suitable as standards for hydroxides/oxides. The reviewer recommends using metal oxides and hydroxides with different oxidation states as standards.

3. The XAS whiteness intensity is influenced by both the oxidation state and the compactness of orbitals. Therefore, using whiteness intensity alone to determine changes in oxidation state is not highly accurate. The reviewer recommends presenting and analyzing the entire XANES spectrum for a more reliable interpretation.

4. The in situ EXAFS data reveal significant changes in M–O distance, intensity, and the appearance/disappearance of the M–M peak (between 2–3 Å) (Figure 5h,k; Figure S27, S28), suggesting a change in oxidation state and structural transformation in the catalysts, likely from hydroxide to oxide/oxyhydroxide. Do the authors have any insights into this transformation and its reversibility?

5. The reviewer recommends including details of the electrochemical and WE tests, such as catalyst loading, scan rate, rotation rate, and iR correction, in the manuscript for clarity.

6. Could the authors clarify why the AEMWE electrolyzer was tested in concentrated KOH (30%) instead of 1 M KOH or lower, given that lower KOH concentrations are less corrosive and a key focus in AEMWE research?

Based on the above concerns, the reviewer believes the manuscript is not suitable for publication in its current form and requires a more careful analysis of the experimental results, particularly the XAS data.

Version 1:

Reviewer comments:

Reviewer #1

(Remarks to the Author)

I appreciate the authors' thorough and careful revision of the manuscript. They have addressed the previous concerns effectively and improved the clarity and quality of the presentation. The additional data and explanations were effective and improved the clarity and quality of the presentation. The additional data and explanations provided have strengthened the arguments and enhanced the overall impact of the study. I have a few additional comments.

1. There is a significant peak shift in the overall O 1s spectra between HE LDH and Na-HE LDH. It would be helpful if the authors could verify this observation and provide a brief explanation.

2. I suggest the authors provide the applied potential/voltage for all chronoamperometry stability graphs.

3. In the comparison table, the authors should consider providing the complete composition instead of abbreviations such as HEG, HEA, MCPS, etc.

4. I suggest the authors add a comparison table for different Na doping levels, including activity, Tafel slope, ECSA, and EIS data.

5. The authors could provide a high-magnification TEM image of the sample (before testing) to clarify the amorphous layer formation during the OER process.

6. I recommend that the authors use a line graph instead of scatter plots for the EPR data presentation.

7. Please include the electrolyte flow rates in the AEMWE section for better reproducibility and clarity.

Reviewer #2

(Remarks to the Author)

The author has carefully revised the comments, and I agree to accept the revised paper.

Reviewer #3

(Remarks to the Author)

The authors have addressed most of my concerns, and I am pleased to recommend acceptance of this manuscript in its current form.

Reviewer #4

(Remarks to the Author)

After revision, the authors have adequately addressed the concerns previously raised, particularly those related to the processing and analysis of the XAS data. The additional details and clarifications provided in both the revised manuscript and the point-by-point response demonstrate a sound understanding of the methodology and ensure that the data analysis is both transparent and scientifically robust. Based on these revisions, the reviewer is satisfied that the XAS data processing and analysis now meet the standards required for publication in this journal.

School of Materials Science and Engineering

Tel: +86-0533-2788087

Shandong University of Technology

Fax: +86-0533-2788087

Zibo 255000, P. R. China

E-mail: conghailin@sdut.edu.cn (H. L. Cong)

May. 13, 2025

Title: Constructing oxygen non-bonding band in high-entropy (oxygen) hydroxides for industrial-scale water oxidation

Dear Editors and Reviewers:

We appreciate your time and efforts in handling our manuscript in *Nature Communications*.

The manuscript has been revised and greatly improved according to reviewers' suggestions and comments. Responses to reviewers' comments are listed as follows.

Thank you very much for your further consideration.

Sincerely yours,

Dr. Hailin Cong

- **Point by point response and action to the comments of Reviewer 1**

Reviewer's general comment:

In this study, the authors reported a Na-doped HE LDH system and investigated the impact of Na on the formation of the oxygen non-bonding band in HE LDH towards enhanced OER activity. The authors conducted various characterizations and electrochemical studies. Significantly, the reported OER overpotential and long-term stability at high current density are interesting. However, addressing the identified issues will further strengthen the manuscript. So, after substantial revisions, the novelty and significance of this work are recommendable for publication in "Nature Communications" journal. The significant concerns of this work are as follows.

Response and action:

Thank you for your pertinent comments. All the comments have been carefully addressed in the revised manuscript. All changes have been highlighted in red in the revised manuscript for your reviewing convenience.

Reviewer's comment (1):

Why did the authors choose a complex system of HE hydroxides to study the formation of the oxygen non-bonding band? This point should be stated clearly.

Answer:

Thank you for your insightful comment regarding our rationale for choosing HE hydroxides as the model system to study the formation of the oxygen non-bonding band (O_{NB}). The selection of HE hydroxides is grounded in both electronic structure requirements and stability considerations essential for O_{NB} formation. The generation of O_{NB} states enables electron removal from non-bonding oxygen rather than the M–O bonding band, thus avoiding structural degradation during lattice oxygen oxidation. However, this process is highly sensitive to the covalency of M–O bonds and the presence of electronic disorder near the Fermi level (Adv. Mater. 2022, 34, 2107956).

*HE hydroxides, with their multi-cationic composition and inherent configurational entropy, offer two key advantages: (1) **Flexible orbital environment:** The coexistence of multiple 3d transition metal cations with different electronegativities and electronic configurations leads to spatially and energetically diverse M–O bonds (Nat. Rev. Mater. 2024, 9, 846–865, Nat. Rev. Mater. 2024, 9, 266–281; Nat. Rev. Chem. 2024, 8, 471–485). This induces local variations in d–p orbital hybridization and Hubbard band splitting, which collectively broaden the O 2p bandwidth and facilitate the emergence of non-bonding oxygen states near the Fermi level. (2) **Enhanced structural robustness:** The high configurational entropy thermodynamically stabilizes the solid solution phase and suppresses long-range cation ordering. This stabilizing effect mitigates metal leaching, suppresses second-phase segregation, and delays amorphization under harsh OER conditions (Nat. Commun. 2023, 14, 6019). This entropy-stabilized matrix ensures that the observed catalytic behavior originates primarily from electronic-level modifications (i.e., Na doping and O_{NB} formation), rather than from irreversible structural degradation. This synergy between high-entropy stabilization and targeted O_{NB} engineering is central to the strategy of our work.*

Reviewer's comment (2):

Carbon cloth is not typically recommended as a substrate for industrial-scale water electrolysis. Why did the authors use it in this study?

Answer:

*Thanks for your reasonable query. It is true that carbon cloth is relatively no resistant to alkali corrosion and tends to dissolve under the harsh conditions of industrial-scale water electrolysis. The initial consideration for using catalyst-loaded carbon cloth for the assembly of AEMWE in this manuscript was that we had conducted a series of structural characterization and electrocatalytic tests using carbon cloth-loaded catalysts, and wanted to maintain the consistency of the tests, which also showed good activity and stability of the Pt/C||Na-HE LDH electrode pair. Considering that nickel foam has very high electrical conductivity and alkali resistance and most of the literature (Energy Environ. Sci., 2024, 17, 5260; Angew. Chem. Int. Ed. 2024, 63, e202407509; Adv. Mater. 2024, 36, 2405970;) also uses nickel foam as the support for assembling AEMWE, we reassembled the AEMWE electrolyzer by loading the catalyst on nickel foam for total hydrolysis and stability tests, and the results are shown in **Supplementary Note 5** and **Fig. S29**. The results show that the AEM water electrolyzer based on Pt/C||Na-HE LDH can achieve high current densities of 100 mA cm⁻², 500 mA cm⁻², and 1000 mA cm⁻² with only 1.51 V, 1.54 V and 1.57 V tank voltages. This is superior to the use of Pt/C||HE LDH and Pt/C||NiFe LDH electrolysis baths (**Fig. S29a-b**). More importantly, the Pt/C||Na-HE LDH electrolyzer also showed excellent long-term stability after 1000 h of continuous testing at ~500 mA cm⁻² (**Fig. S29c**). This confirms the potential and advantages of Pt/C||Na-HE LDH for large-scale industrial hydrogen production.*

Actions:

- **Action 1, on Page 11, Line 297 of the revised manuscript**

Added statement:

“Given the high electrical conductivity and alkali resistance of nickel foam (NF), we coated the catalyst on NF to assemble the AEMWE, and the Pt/C||Na-HE LDH electrode pair still maintains high activity and stability, as detailed in **Supplementary Note 5** and **Fig. S29**.”

- **Action 2, on Page 37 of the Supplementary Information**

Added note:

“Supplementary Note 5. Investigate the AEMWE properties of Na-HE LDH on nickel foam (NF) substrates.

Considering the high electrical conductivity and alkali resistance of nickel foam, we also reassembled the AEMWE by loading the catalyst on NF for activity and stability tests. As shown in **Fig. S29a-b**, AEMWE based on Pt/C||Na-HE LDH achieves high current densities of 100 mA cm⁻², 500 mA cm⁻² and 1000 mA cm⁻² at cell voltages as low as 1.51 V, 1.54 V

and 1.57 V. This is superior to the use of Pt/C||HE LDH and Pt/C||NiFe LDH electrolysis cells. Here we note that due to the high conductivity of NF, AEMWE with NF as the carrier has higher activity than AEMWE with carbon cloth as the carrier, especially at high current densities. More importantly, the Pt/C||Na-HE LDH electrolyser with NF as the carrier also showed excellent long-term stability after 1000 h of continuous testing at $\sim 500 \text{ mA cm}^{-2}$ (Fig. S29c).”

• **Action 3, on Page 38 of the Supplementary Information**

Added Figure:

Fig. S29 (a) LSV curves of Pt/C|| Na-HE LDH, Pt/C||HE LDH and Pt/C||NiFe LDH in 30 wt% KOH at 60°C. (b) Comparison of voltage values for Pt/C|| Na-HE LDH, Pt/C||HE LDH and Pt/C||NiFe LDH at different current densities. (c) i-t measurement for Pt/C|| Na-HE LDH, Pt/C||HE LDH and Pt/C||NiFe LDH at 1.55 V vs. RHE, 1.65 V vs. RHE and 1.61 V vs. RHE.

Reviewer's comment (3):

What do the authors mean by "industrial-scale electrolysis"? A 4 cm² area is generally not considered industrial scale. Could the authors use a larger electrolyzer to support their claim better? Additionally, how was the catalyst loading of 1 mg/cm² determined?

Answer:

Thanks for your insightful comment. In our manuscript, the term "industrial-scale electrolysis" is not meant to refer strictly to the geometric area of the electrode, but rather to the reaction conditions that closely simulate those found in practical industrial applications. Specifically, our electrolyzer operates at a high current density of 500 mA cm⁻², in concentrated 30 wt% KOH, at elevated temperature (60 °C), and maintains stable performance for over 2000 hours. These conditions are widely accepted in literature as representative of industrially relevant testing. Indeed, many recent works use electrode areas of 1 cm² and 4 cm² while still referring to "industrial" conditions based on electrochemical load rather than size alone (Nat. Commun. 2025, 16, 215; Nat. Commun. 2025, 16, 3502; Nat. Commun. 2025, 16, 2908).

The catalyst loading of 1 mg cm⁻² was determined by weighing the carbon cloth substrate before and after catalyst deposition and calculating the mass difference. This process was repeated three times on independently prepared samples, and the average value was reported. We have now added the detailed loading data to Table S4 in the Supporting Information.

Actions:

- **Action 1, on Page 64 of the Supplementary Information**

Added table:

Table S4. The loading amounts of Na-HE LDH and HE LDH.

Sample	Loading amounts (mg cm ⁻²)			
	1 st	2 nd	3 rd	Mean
Na-HE LDH	0.98	0.94	1.12	1.01
HE LDH	0.93	1.05	1.02	1.00

Reviewer's comment (4):

Have the authors conducted a cost analysis of their system? How does it compare to conventional systems of the same scale?

Answer:

Thank you for the helpful suggestion. We have added a cost analysis of the Na-HE LDH catalyst, including the estimated raw material cost based on market prices. For comparison, we also included the cost of commercial IrO₂. This analysis demonstrates the significant economic advantage of our catalyst while maintaining excellent performance and durability under industrial conditions. Specifically, the estimated material cost per batch of Na-HE LDH is 0.145 \$ mg⁻¹, which is substantially lower than that of IrO₂ (0.702 \$ mg⁻¹).

Actions:

- **Action 1, on Page 11, Line 301 of the revised manuscript**

Added statement:

“In the case of industrial alkaline water electrolysis, the use of Na-HE LDH electrocatalysts is advantageous because it uses a large number of highly abundant and low-cost elements (Table S6). Considering the estimated manufacturing cost of the catalyst, which totaled 0.145 \$ mg⁻¹, significantly lower than current commercial catalysts such as IrO₂ (0.702 \$ mg⁻¹), Na-HE LDH emerged as a promising alternative to precious metal catalysts.”

- **Action 2, on Page 71 of the Supplementary Information**

Added table:

Table S6. Cost analysis of Na-HE LDH.

	Reagent	Unit Price (USD/kg)	Used Amount (mmol)	Used Amount (g)	Cost per Batch (USD)
Na- HE LDH	Ni(NO ₃) ₂ ·6H ₂ O	295	0.45	0.1309	0.0386
	Fe(NO ₃) ₂ ·9H ₂ O	191	0.45	0.1818	0.0347
	Co(NO ₃) ₂ ·6H ₂ O	564	0.45	0.1310	0.0739
	Mn(NO ₃) ₂ ·4H ₂ O	244	0.45	0.1130	0.0276
	Cu(NO ₃) ₂ ·3H ₂ O	408	0.45	0.1087	0.0443
	NH ₄ F	472	4	0.1482	0.0699
	urea	121	10	0.6006	0.0727
	NaCl	62	0.045	0.0026	0.0002
	Carbon cloth	14/(16*16 cm)	-	2*2 cm	0.2188
	Total	-	-	4 mg	0.5807
IrO ₂	-	\$702/g	-	10 mg	7.02

Note: All price information is from Sigma-Aldrich, as of April 19, 2025.

Reviewer's comment (5):

What is the thickness of the nanosheets based on SEM and TEM analysis? I recommend that the authors provide low-magnification SEM and TEM images to confirm the uniformity of the nanosheets on each carbon fiber in the CC support.

Answer:

Thank you for your suggestion. The thickness of the nanosheets of Na-HE LDH measured by atomic force microscopy (AFM) is 4.99 nm. we have again tested the thickness of the nanosheets shown in the SEM, and the results show that the thickness of the nanosheets of Na-HE LDH is 5.68 nm (**Fig. R1**), which is almost the same as the AFM results. Since the side of the nanosheets cannot be observed in the TEM image, we were not able to measure the thickness of the nanosheets in the TEM. In addition, we also provide low and high magnification SEM images (**Fig. S5**), which show that Na-HE LDH grows uniformly on the carbon fiber.

Fig. R1 Thickness distribution of Na-HE LDH nanosheets according to SEM.

Actions:

- **Action 1, on Page 10 of the Supplementary Information**

Added figure:

Fig. S5. SEM images of Na-HE LDH at different magnifications.

Reviewer's comment (6):

Why didn't the authors include oxygen TEM elemental mapping? The TEM elemental mapping of Na atoms is unclear in all the provided images. I recommend that the authors present more explicit images with more distinguishable colors.

Answer:

*Thank you for the reminder. We have added the elemental map of O as **Fig. S7**, where the O element is uniformly distributed on the nanosheet. In addition, we adjusted the image contrast to provide clearer images (**Fig. 2h**) for reviewers and readers.*

Actions:

- **Action 1, on Page 5, Line 117 of the revised manuscript**

Revised figure:

Fig. 2h EDS-Mapping of Na-HE LDH.

- **Action 2, on Page 12 of the Supplementary Information**

Added figure:

Fig. S7 EDS-Mapping of O of Na-HE LDH.

Reviewer's comment (7):

The authors should provide more detailed information about the AEMWE assembly and its conditions.

Answer:

Thank you for your reminder. We have detailed experimental section related to cell evaluation in Assembly of anion exchange membrane water electrolysis and Materials sections of the revised manuscript.

Actions:

- **Action 1, on Page 21, Line 552 of the revised manuscript**

Added discussion:

“4.5 Assembly of anion exchange membrane water electrolysis.

The anion exchange membrane water electrolyze system consists of four components: power supply, temperature control system, flow control system, and anion exchange membrane water solution. The electrolyser consists of two corrosion-resistant TA1 grade plates with SUS 316 end plates and a curved flow field machined into the bipolar plates. The anode and cathode electrodes are separated by an anion-exchange membrane to prevent gas crossings and are pressed against the sides of the electrolyze to form a membrane electrode assembly (MEA). The MEA was made of an anion exchange membrane (FAA-3-50) with cathode (Pt/C, an area of 4 cm² and a loading of 1 mg cm⁻²) and anode (Na-HE LDH, an area of 4 cm² and a loading of 1 mg cm⁻²) catalysts loaded on a carbon cloth and located on both sides of the membrane. The temperature of the electrolytic cell was maintained at 60° C and the electrolyte was 30 wt% KOH.”

Reviewer's comment (8):

How about the performance and stability of the Na-HE LDH in other electrolytes?

Answer:

*Usually, the alkaline electrolytes are 1 M KOH and 1 M NaOH. in this manuscript, we used 1 M KOH solution and we also added the performance and stability of Na-HE LDH in 1 M NaOH. As shown in **Fig. S27**, the overpotential of Na-HE LDH in 1 M NaOH is 204 mV, Tafel slope is 38.5 mV dec⁻¹, C_{dl} value is 3.9 mF cm⁻², and charge transfer resistance is 3.4 Ω. This is similar to the performance of Na-HE LDH in 1 M KOH. We also tested the long-term stability of Na-HE LDH in 1 M NaOH, and the results show that Na-HE LDH still shows super-long durability, with almost no decay after 1000 h continuous operation. In conclusion, Na-HE LDH has excellent OER activity and stability in both alkaline electrolytes.*

Actions:

- **Action 1, on Page 9, Line 244 of the revised manuscript**

Added discussion:

“Moreover, the OER performance of Na-HE LDH in 1 M NaOH electrolyte was also explored, see **Supplementary Note 4** and **Fig. S27**.”

- **Action 2, on Page 34 of the Supplementary Information**

Added note:

“Supplementary Note 4. Investigate the OER activity and stability of Na-HE LDH in 1 M NaOH electrolyte.

To confirm the robust activity and stability of Na-HE LDH in an alkaline environment, we again tested it with 1 M NaOH as the electrolyte. As shown in **Fig. S27**, the overpotential of Na-HE LDH in 1 M NaOH is 204 mV, the Tafel slope is 38.5 mV dec⁻¹, the C_{dl} value is 3.9 mF cm⁻² and the charge transfer resistance is 3.4 Ω, which is similar to the performance of Na-HE LDH in 1 M KOH. We also tested the long-term stability of Na-HE LDH in 1 M NaOH, and the results showed that Na-HE LDH still has an extremely long durability with almost no decay after 1000 h of continuous operation. In conclusion, Na-HE LDH has excellent OER activity and stability in different alkaline electrolytes and has great potential for use.”

- **Action 3, on Page 35 of the Supplementary Information**

Added figure:

Figure S27. OER performance of Na-HE LDH OER performance in 1 M NaOH. (a) LSV curve, (b) Tafel plot, (c) C_{dl} plot, (d) EIS curve and (e) stability test.

Reviewer's comment (9):

The authors recommend comparing their performance with other high-entropy systems of similar composition in the main manuscript, such as Surfaces and Interfaces 46 (2024) 104084, Adv. Funct. Mater. 2023, 33, 2301153, J. Mater. Chem. A, 2021, 9, 16841-16851, Nat Commun 14, 6019 (2023).

Answer:

Thank you for your suggestions. These are high level papers on high entropy materials and we have compared the high entropy materials of above papers with the Na-HE LDH in this manuscript, listed in **Table S3**. Moreover, we have also cited the above articles in the revised manuscript.

Actions:

- **Action 1, on Page 63 of the Supplementary Information**

Added table:

Table S3. Comparison of OER performances of Na-HE LDH with previously reported well-performed OER electrocatalysts.

Catalyst	$\eta@$ 10mA cm ² (mV)	$\eta@$ 50 mA cm ² (mV)	$\eta@$ 100 mA cm ² (mV)	Electrolyte	Ref
Na-HE LDH	176	228	263	1M KOH	This work
HEO/MWCNT	350	-		1M KOH	1
CV-activated MnFeCoNi HEA	302	-		1M KOH	2
MCPS	288	-		1M KOH	3
HF-HEA	265	-		1M KOH	4
Fe-Cr-Co-Ni-Cu HE- LDHs-Ar-20	330	-		1M KOH	4
M ₃ O _{3.2} HEO	336	-		1M KOH	5
L5M2Co	380	-		1M KOH	6
HEG	229	-		1M KOH	7
NiCoFeCrMo-based HEH	292	-		1M KOH	8
FeCoNiMn	266	-		1M KOH	9
FeNiCoAl	225	-		1M KOH	10
HEA-60h		479	-	1M KOH	11
V _{1.0} -HEA		370	-	1M KOH	12
FeCoNiCuMn@CF HEAs			260	1M KOH	13
Au _{SA} -MnFeCoNiCu LDH	213		260	1M KOH	14

Reviewer's comment (10):

Did the authors conduct any control experiments on the Na doping by varying the Na content?

Answer:

*Thank you for your reminder. We have previously performed control experiments with Na doping. The samples with optimum Na doping are shown in the original manuscript. The results of the performance tests with different Na doping levels have been further added to the revised manuscript. As shown in **Supplementary Note 1** and **Fig. S1**, compared with Na_{0.025}-HE LDH*

(232 mV@10 mA cm⁻²) and Na_{0.065}-HE LDH (277 mV@10 mA cm⁻²), Na_{0.045}-HE LDH (176 mV@10 mA cm⁻²) possesses the lowest overpotential, the fastest reaction kinetics, the largest electrocatalytically active surface area and the smallest charge transfer resistance. Therefore, we used Na_{0.045}-HE LDH (labelled as Na-HE LDH in the manuscript) as the optimal sample to compare with pristine HE LDH to investigate the intrinsic mechanism of Na on the LOM mechanism of activating HE LDH.

Actions:

- **Action 1, on Page 4, Line 100 of the revised manuscript**

Added discussion:

“If not otherwise stated, all of the following Na-HE LDH refer to Na_{0.045}-HE LDH, this is due to its excellent OER properties, see **Supplementary Note 1** for details.”

- **Action 2, on Page 5 of the Supplementary Information**

Added note:

“Supplementary Note 1. Investigate the effect of Na doping on the OER activity of HE LDH.

To investigate the effect of different Na doping amounts on the OER activity of HE LDH, we performed electrochemical performance tests on Na_{0.025}-HE LDH, Na_{0.045}-HE LDH and Na_{0.065}-HE LDH, and the results are shown in **Fig. S1**. Na_{0.045}-HE LDH exhibited the most excellent OER activity, with an over potential of 176 mV at 10 mA cm⁻² (**Fig. S1a**), which is lower than Na_{0.025}-HE LDH (232 mV) and Na_{0.065}-HE LDH (277 mV). In addition, Na_{0.045}-HE LDH exhibits the fastest reaction kinetics (**Fig. S1b**), the largest electrocatalytically active surface area (**Fig. S1c**) and the smallest charge transfer resistance (**Fig. S1d**). Too little Na doping may lead to incomplete lattice oxygen activation and thus limited enhancement of OER activity. Too much Na doping may lead to a relative decrease in active sites. Therefore, Na_{0.045}-HE LDH (labelled as Na-HE LDH in the manuscript) was chosen as the target catalyst to compare with pristine HE LDH to investigate the intrinsic mechanism of Na ions on the activation of HE LDH.”

- **Action 3, on Page 20 Line 499 of the revised manuscript**

Added discussion:

“4.1 Preparation of HE LDH and Na-HE LDH.

The HE LDH was synthesized using a hydrothermal approach. Initially, 0.45 mmol each of Ni(NO₃)₂·6H₂O, Fe(NO₃)₂·9H₂O, Co(NO₃)₂·6H₂O, Mn(NO₃)₂·4H₂O, and Cu(NO₃)₂·3H₂O

were dissolved in 35 mL of deionized water. To this solution, 4 mmol of NH_4F and 10 mmol of urea were added. After ensuring complete dissolution, the mixture was transferred into an autoclave, along with a pre-treated piece of carbon cloth (CC), and heated at 120°C for 6 h. $\text{Na}_{0.045}\text{-HE LDH}$ (labelled as Na-HE LDH in the text) was obtained by adding 0.045 mmol NaCl to the precursor, otherwise consistent with the above steps. In addition, $\text{Na}_{0.025}\text{-HE LDH}$ and $\text{Na}_{0.065}\text{-HE LDH}$ were obtained by adding 0.025 mmol NaCl and 0.065 mmol NaCl to the precursor.”

• **Action 4, on Page 6 of the Supplementary Information**

Added figure:

Fig. S1 OER performance of $\text{Na}_{0.025}\text{-HE LDH}$, $\text{Na}_{0.045}\text{-HE LDH}$ and $\text{Na}_{0.065}\text{-HE LDH}$ in 1.0 M KOH. (a) LSV curves, (b) Tafel plots, (c) C_{dl} plots and (d) EIS curves.

Reviewer's comment (11):

Interestingly, the oxidation states of Mn, Fe, Co, and Cu remain unchanged while Ni undergoes significant oxidation during OER. How do the authors explain it?

Answer:

We confirmed the changes in elemental valence states by recording the intensity changes of the white line peaks at different potentials in the in situ XAS spectra. As shown in **Figs. 5g and j**, the valence states of Mn, Fe, Co, and Cu in Na-HE LDH and HE LDH increased slightly with increasing applied voltage, while the valence states of Ni in Na-HE LDH and HE LDH increased significantly, which suggests that Ni strongly participates in the OER reaction, indicating that Ni is an active site in the AEM pathway (*Angew. Chem. Int. Ed.* 2024, 63, 202407509; *Nat Catal.* 2021, 4, 212–222; *J. Am. Chem. Soc.* 2023, 145, 23659–23669). It is important to note here that although the LOM mechanism is O as the active site, there is also a part of the AEM mechanism in the whole OER, and Ni is the active site in the AEM mechanism, which leads to the drastic change of Ni valence.

Fig. 5 Changes in the normalized absorbance of 3d transitional elements in the in-situ K-edge XANES spectra of (g) HE LDH and (j) Na-HE LDH

Actions:

- **Action 1, on Page 15 Line 380 of the revised manuscript**

Revised discussion:

“The in situ XANES spectra show that the valence states of Ni in Na-HE LDH and HE LDH increase significantly from the open-circuit potential (OCV) to 1.7 V (**Fig. 5g** and **Fig. 5j**) whereas the valence states of Mn, Fe, Co, Ni and Cu do not change much (**Fig. S37-S38**). We further give the values of valence changes with increasing voltage for all metallic elements (**Figs. S39-S40**), where the valence of all the metal elements increases with increasing oxidation potential, suggesting that the HE LDH and Na-HE LDH are reconstruction. It is noted that among all the elements of HE LDH, the Ni has the largest

change in valence state (0.45, from +2.71 to +3.16), and the Ni also has the largest change in valence in Na-HE LDH (0.48, from +2.77 to +3.25). This indicates that Ni involved in the OER process through electron transfer and is evaluated as the active site. It is important to note here that although the LOM mechanism is O as the active site, there is also a part of the AEM mechanism in the whole OER, and Ni is the active site in the AEM mechanism, which leads to the drastic change of Ni valence. It is noteworthy that the valence state of Ni is higher in Na-HE LDH compared to HE LDH, which is more favorable to the OER process, consistent with the in-situ Raman results.”

Reviewer’s comment (12):

What is the influence of the amorphous layer on the long-term catalytic performance of Na-HE LDH?

Answer:

Thanks for raising this important question. As observed in our work, a thin amorphous layer (~20 nm) forms on the surface of Na-HE LDH after long-term electrolysis. Notably, despite the presence of this amorphous layer, the catalyst exhibits remarkable durability, maintaining stable operation over 2000 h at 500 mA cm⁻² with only minimal degradation. This observation suggests that the amorphous layer does not deteriorate the catalytic performance.

Instead, we propose that the amorphous layer acts as a protective barrier that suppresses further corrosion, stabilizes surface oxygen vacancies, and prevents excessive metal leaching. Similar protective effects of surface amorphization have been reported in recent studies. For instance, the formation of an amorphous hydroxide layer has been shown to terminate uncontrolled structural reconstruction and enhance catalyst durability in alkaline OER (Nat. Commun. 2025, 16, 3407). Furthermore, operando studies have demonstrated that appropriately controlled surface restructuring, including partial amorphization, can mitigate irreversible degradation and promote reliable high-current electrolysis (Nat. Mater. 2025, 24, 1). Additionally, structural modifications, such as forming dense interlayers or amorphous-like transition regions, have been proven effective in enhancing catalyst stability under industrial-scale operating conditions (Nat. Catal. 2024, 7, 944).

Accordingly, based on our experimental results and supported by these findings, we believe that the amorphous layer generated during long-term operation contributes positively by providing a stable and protective surface, thereby ensuring the outstanding durability of the Na-HE LDH catalyst.

- **Point by point response and action to the comments of Reviewer 2**

Reviewer's general comment:

It is my pleasure to participate in the peer review of this manuscript for Nature Communications. The article provides a comprehensive analysis and mechanistic interpretation of a sodium-incorporated high-entropy Mn-Fe-Co-Ni-Cu hydroxide for alkaline water electrolysis. The material demonstrates exceptional performance with an overpotential of 176 mV at 10 mA cm⁻² and a remarkable durability of 2000 hours at a current density of 500 mA cm⁻². The study proposes that the oxygen evolution follows the lattice oxygen mechanism (LOM) with oxygen sites adjacent to Ni centers acting as active sites, supported by thorough mechanistic investigations. While this work shows promise in addressing the stability challenges typically associated with LOM systems, as a reviewer, I would like to raise several critical questions and observations to ensure scientific rigor.

Response and action:

Thank you for your pertinent comments. All the comments have been carefully addressed in the revised manuscript. All changes have been highlighted in red in the revised manuscript for your reviewing convenience.

Reviewer's comment (1):

Has the impact of adding different sodium concentrations on performance been explored to obtain the optimal sodium concentration? The process of exploring the optimal concentration needs to be seen in the supporting materials.

Answer:

*Thank you for your reminder. We have previously performed controlled experiments with Na doping. The samples with optimum Na doping are shown in the original manuscript. The results of the performance tests with different Na doping levels have been added to the revised manuscript. As shown in **Supplementary Note 1** and **Fig. S1**, compared with Na_{0.025}-HE LDH (232 mV@10 mA cm⁻²) and Na_{0.065}-HE LDH (277 mV@10 mA cm⁻²), Na_{0.045}-HE LDH (176 mV@10 mA cm⁻²) possesses the lowest overpotential, the fastest reaction kinetics, the largest electrocatalytically active surface area and the smallest charge transfer resistance. Therefore, we used Na_{0.045}-HE LDH (labelled as Na-HE LDH in the manuscript) as the optimal sample to compare with pristine HE LDH to investigate the intrinsic mechanism of Na on the LOM mechanism of activating HE LDH.*

Actions:

- **Action 1, on Page 4, Line 100 of the revised manuscript**

Added discussion:

“If not otherwise stated, all of the following Na-HE LDH refer to Na_{0.045}-HE LDH, this is due to its excellent OER properties, see Supplementary Note 1 for details.”

- **Action 2, on Page 5 of the Supplementary Information**

Added discussion:

“Supplementary Note 1. Investigate the effect of Na doping on the OER activity of HE LDH.

To investigate the effect of different Na doping amounts on the OER activity of HE LDH, we performed electrochemical performance tests on Na_{0.025}-HE LDH, Na_{0.045}-HE LDH and Na_{0.065}-HE LDH, and the results are shown in **Fig. S1**. Na_{0.045}-HE LDH exhibited the most excellent OER activity, with an over potential of 176 mV at 10 mA cm⁻² (**Fig. S1a**), which is lower than Na_{0.025}-HE LDH (232 mV) and Na_{0.065}-HE LDH (277 mV). In addition, Na_{0.045}-HE LDH exhibits the fastest reaction kinetics (**Fig. S1b**), the largest electrocatalytically active surface area (**Fig. S1c**) and the smallest charge transfer resistance (**Fig. S1d**). Too little Na doping may lead to incomplete lattice oxygen activation and thus limited enhancement of OER activity. Too much Na doping may lead to a relative decrease in active sites. Therefore, Na_{0.045}-HE LDH (labelled as Na-HE LDH in the manuscript) was chosen as the target catalyst to compare with pristine HE LDH to investigate the intrinsic mechanism of Na ions on the activation of HE LDH.”

- **Action 3, on Page 20, Line 499 of the revised manuscript**

Added discussion:

“4.1 Preparation of HE LDH and Na-HE LDH.

The HE LDH was synthesized using a hydrothermal approach. Initially, 0.45 mmol each of Ni(NO₃)₂·6H₂O, Fe(NO₃)₂·9H₂O, Co(NO₃)₂·6H₂O, Mn(NO₃)₂·4H₂O, and Cu(NO₃)₂·3H₂O were dissolved in 35 mL of deionized water. To this solution, 4 mmol of NH₄F and 10 mmol of urea were added. After ensuring complete dissolution, the mixture was transferred into an autoclave, along with a pre-treated piece of carbon cloth (CC), and heated at 120°C for 6 h. Na_{0.045}-HE LDH (labelled as Na-HE LDH in the text) was obtained by adding 0.045 mmol NaCl to the precursor, otherwise consistent with the above steps. In addition, Na_{0.025}-HE LDH and Na_{0.065}-HE LDH were obtained by adding 0.025 mmol NaCl and 0.065 mmol NaCl to the precursor.”

- **Action 4, on Page 6 of the Supplementary Information**

Added figure:

Fig. S1 OER performance of Na_{0.025}-HE LDH, Na_{0.045}-HE LDH and Na_{0.065}-HE LDH in 1.0 M KOH. (a) LSV curves, (b) Tafel plots, (c) C_{dl} plots and (d) EIS curves.

Reviewer's comment (2):

The full spectrum of XPS analysis should be displayed in the material.

Answer:

Thanks to your reminder, we have added the XPS survey spectra of Na-HE LDH and HE LDH in the revised supporting information. The results (**Fig. S10a**) show that both Mn, Fe, Co, Ni, Cu and O elements are present. In addition, the element Na is present in Na-HE LDH.

Actions:

- **Action 1, on Page 15 of the Supplementary Information**

Added figure:

Fig. S10. (a) Survey XPS spectra of Na-HE LDH and HE LDH.

Reviewer's comment (3):

The reasons for adding higher concentrations of NH_4F and urea during the preparation process and their respective effects.

Answer:

We sincerely thank the reviewer for pointing out the need to clarify the role of NH_4F and urea in our synthesis. We have now added the descriptions in Method section of the revised manuscript to elucidate the respective functions of these two components.

NH_4F plays a critical role as a structural modulator, facilitating the formation of LDH nanosheets and controlling the surface morphology. Fluoride ions derived from NH_4F can coordinate with transition metal ions to form $[\text{MF}_x]$ complexes, which are hydrolyzed during the reaction to guide anisotropic crystal growth. The work by Xing et al. (RSC Adv., 2017, 7, 38945) provides direct evidence that NH_4F addition enables the formation of unique nanowire or nanosheet morphologies in Ni–Co-LDHs, and significantly enhances the crystallinity, surface area, and conductivity of the final product. Moreover, NH_4F contributes to better adhesion of LDH arrays to the substrate, ensuring higher structural stability and loading.

Urea serves as a homogeneous and gradual OH^- source via thermal decomposition during the hydrothermal process. This slow release of hydroxide ions promotes controlled nucleation and growth of highly crystalline LDH structures, as demonstrated in previous studies using the urea method for layered hydroxide synthesis (Dalton Trans., 2024, 53, 12543). For our HE LDH, the use of relatively high urea concentration ensures sufficient alkalinity for the co-precipitation of multiple metal cations, which is essential for the uniform formation of LDH with tailored morphology.

Collectively, in our case, the combined use of urea and NH_4F was systematically optimized to achieve ultrathin 2D HE LDH nanosheets with a uniform thickness of ~ 3.7 nm (4 layers), as confirmed by AFM and TEM. These optimized conditions were critical to achieving high OER activity under industrial current densities.

Actions:

- **Action 1, on Page 20, Line 502 of the revised manuscript**

Added discussion:

“To this solution, 4 mmol of NH_4F and 10 mmol of urea were added, which serve to control the nucleation rate and regulate the nanosheet morphology.”

Reviewer’s comment (4):

In Figure 4k, it should be indicated in the graph that the change in long-term test is at 1.7 V. The current annotation should be incorrect and difficult to understand. And this testing method for controlling voltage is somewhat inconsistent with the previous statement that the control current remains unchanged after 2000 hours of continuous operation at 500 mA cm^{-2} .

Answer:

Thank you for the reminder. Fig. 4k shows the results measured at a fixed voltage with an initial current density of about 500 mA cm^{-2} . We have removed the annotations from Fig. 4k and modified the presentation of the manuscript and figure notes. Thank you again for your careful scrutiny.

Actions:

- **Action 1, on Page 11, Line 295 of the revised manuscript**

Added discussion:

“More importantly, the Pt/C||Na-HE LDH electrolyzer also exhibits excellent long-term stability when tested continuously for 2000 h at ~ 500 mA cm^{-2} (Fig. 4k).”

- **Action 2, on Page 13, Line 316 of the revised manuscript**

Added discussion:

“Fig. 4k i-t measurement for Pt/C|| Na-HE LDH, Pt/C||HE LDH and Pt/C||NiFe LDH at 1.56 V vs. RHE, 1.71 V vs. RHE and 1.69 V vs. RHE.”

Reviewer’s comment (5):

Can you conduct more in-depth research on the amorphous layer generated on the surface of the material, such as capturing its structure through transmission electron microscopy?

Answer:

Thank you for your suggestion. The in-situ Raman results show that Na-HE LDH produced an amorphous layer on the surface during OER and the broad peaks are attributed to $\text{M}^{\text{III}}\text{-O}$ peaks (Fig. 5c). We added the HRTEM characterization the Na-HE LDH after OER (Fig. S32), and identified the surface amorphous layer to be about 20 nm. further mapping tests on the Na-HE LDH after OER revealed that the Mn, Fe, Co, Ni, Cu and Na elements were still

uniformly distributed without obvious aggregation and segregation (**Fig. S33**). The above results confirm that the amorphous layer generated on the surface of Na-HE LDH is HEOOH (*Na-MnFeCoNiCuOOH*).

Fig. 5c In-situ Raman spectra for Na-HE LDH.

Fig. S33 EDS-Mapping of Na-HE LDH after long-term OER test.

Actions:

- **Action 1, on Page 41 of the Supplementary Information**

Added figure:

Fig. S32 HRTEM image of Na-HE LDH after long-term OER test.

Reviewer's comment (6):

The author states that O near Ni acts as the active site, while when calculating the conversion frequency TOF, n is the number of moles of Ni on the electrode. Is this unreasonable?

Answer:

*Thank you for the reminder that this was an oversight on our part. We have re-compared the TOF of the two samples, where HE LDH is with Ni as the active site and Na-HE LDH is with O as the active site. The results are shown in the revised **Fig. S21**, where the turnover frequency (TOF) of Na-HE LDH at $\eta = 250$ mV is 0.052 s⁻¹, which is 14.4 times higher than that of HE LDH (0.0036 s⁻¹). The above results indicate that the intrinsic activity of HE LDH can be significantly increased by introducing the low valent metal Na.*

Actions:

- **Action 1, on Page 26 of the Supplementary Information**

Added figure:

Fig. S21 Turnover frequency (TOF) at 250 mV for Na-HE LDH and HE LDH. O is the active site in Na-HE LDH and Ni is the active site in HE LDH.

- **Action 2, on Page 9, Line 220 of the revised manuscript**

Revised discussion:

“Similarly, the turnover frequency (TOF, **Fig. S21**) of Na-HE LDH at $\eta=250$ mV is 0.052 s⁻¹, which is 14.4 times that of HE LDH (0.0036 s⁻¹).”

- **Action 3, on Page 21, Line 539 of the revised manuscript**

Revised discussion:

“Turnover frequency (TOF) calculations:

$$\text{TOF} = (j \times A) / (2 \times n \times F)$$

where j is the measured current density, A is geometric area of the electrode, F is the Faraday constant (96485 C mol⁻¹). n is the number of moles of Ni on the HE LDH and O on the Na-HE LDH.”

- **Point by point response and action to the comments of Reviewer 3**

Reviewer's general comment:

Wang et al. propose an ONB engineering strategy to address structural degradation in LOM-mediated reactive hydroxides. The synthesized ortho-hexagonal Na-HE LDH demonstrates promising electrocatalytic performance, achieving an overpotential of 176 mV at 10 mA cm⁻² and maintaining stability for 2000 hours at 500 mA cm⁻² in AEMWE applications. In addition, the contributions of LOM for Na-HE LDH have been quantified by TMAOH. While the authors executed comprehensive structural characterization protocols, the discussion section lacks atomic-scale mechanistic insights into how the introduced ONB features govern electrocatalytic. Overall, while the experimental characterization demonstrates technical competence, the work suffers from the superficial interpretation of characterization data and lacks hierarchical analysis bridging nano/microstructural features to macroscopic catalytic performance. I therefore recommend rejection in its current form.

Response and action:

Thank you for your pertinent comments. All the comments have been carefully addressed in the revised manuscript. All changes have been highlighted in red in the revised manuscript for your reviewing convenience.

Reviewer's comment (1):

The authors need to explain the underlying considerations of the selection of HE LDH, is the high-entropy configuration truly essential? Could the performance improvement be solely attributed to Na doping rather than the high-entropy architecture? The independent contributions of these two factors lack sufficient experimental validation.

Answer:

*Thanks to your valuable comments, we have added experimental verification of the necessity of selecting high-entropy LDH for OER stability and Na doping for catalyst activity enhancement. The corresponding descriptions and experimental data are added in **Supplementary Note 2** and **Supplementary Note 3** of the revised supporting information.*

Actions:

- **Action 1, on Page 21 of the Supplementary Information**

Added note:

“Supplementary Note 2. Investigate the effect of Na doping on the OER activity of conventional binary LDH.

To elucidate the reason for the improved performance, we prepared conventional binary LDHs (NiFe LDH, NiCo LDH, NiCu LDH, and NiMn LDH) and Na-doped binary LDHs (Na-NiFe LDH, Na-NiCo LDH, Na-NiCu LDH, and Na-NiMn LDH) and compared them

with the HE LDH and Na-HE LDH (**Fig. S16**). The LSV curves (**Fig. S17**) show that Na doping (Na-NiFe LDH (223 mV), Na-NiCo LDH (284 mV), Na-NiCu LDH (317 mV) and Na-NiMn LDH (326 mV)) significantly enhances the OER activities of the conventional binary LDHs (NiFe LDH (234 mV), NiCo LDH (307 mV), NiCu LDH (323 mV) and NiMn LDH (337 mV)). Whereas, the activity of HE LDH (330 mV, **Fig. 4a**) is lower than that of conventional binary LDH. this suggests that the improved performance is due to the introduction of Na ions and not due to the high entropy structure.”

- **Action 2, on Page 22 of the Supplementary Information**

Added figure:

Fig. S16 XRD patterns of NiMn LDH, NiFe LDH, NiCo LDH, NiCu LDH, Na-NiMn LDH, Na-NiFe LDH, Na-NiCo LDH and Na-NiCu LDH.

- **Action 3, on Page 23 of the Supplementary Information**

Added figure:

Fig. S17 LSV curves of NiMn LDH, NiFe LDH, NiCo LDH, NiCu LDH, Na-NiMn LDH, Na-NiFe LDH, Na-NiCo LDH and Na-NiCu LDH.

- **Action 4, on Page 33 of the Supplementary Information**

Added note:

“**Supplementary Note 3. Exploring the importance of high entropy structures for OER stability.**

High-entropy structures have high stability due to high-entropy effects and hysteretic diffusion effects, which are essential for catalysts to sustainably undergo OER (*Nat. Commun.* 2023,14, 6019; *Nat. Rev. Mater.* 2024, 9, 266-281). To elucidate the necessity of high-entropy structures, we performed stability tests on conventional binary LDHs and compared them with HE LDHs. The results are shown in **Fig. S26**, where NiFe LDH was continuously stable at $\sim 500 \text{ mA cm}^{-2}$ for 55 h, NiCo LDH, NiCu LDH, NiMn LDH were rapidly deactivated at the early stage of the stability test, whereas HE LDH was continuously operated at $\sim 500 \text{ mA cm}^{-2}$ for 1000 h. This confirmed the necessity of high-entropy structure for long-term OER, especially based on the LOM pathway of OER.”

- **Action 5, on Page 34 of the Supplementary Information**

Added figure:

Fig. S26 Stability test of NiFe LDH, NiCu LDH, NiCo LDH, NiMn LDH and NiCu LDH.

Reviewer's comment (2):

The authors only detected the vibrational modes of Ni-O species in the Raman spectrum, can you explain that? The layer number reduction (5 → 4) induced by Na incorporation in HE LDHs necessitates mechanistic interpretation.

Answer:

Thank you for the valuable comments. Firstly, for the Raman spectra, in our high-entropy (oxy)hydroxide systems containing multiple transition metals such as Ni, Fe, Co, Mn, and Cu, the M-O vibrational modes typically fall within a similar wavenumber range (approximately 400-600 cm^{-1}), leading to significant spectral overlap and making it challenging to distinguish individual contributions in the Raman spectrum. As a result, many studies on similar Ni-based layered double hydroxides or high-entropy compositions adopt the characteristic Ni-O vibrations as representative Raman features, even in the presence of other metal species. This approach is consistently observed across multiple reports involving NiFe-LDH or NiFe-based heterostructures, where the Raman analysis focuses on peaks attributable to Ni-O while other M-O vibrations are either not resolved or not assigned separately (Nat. Commun., 2025, 16, 3407; Nat. Commun., 2024, 15, 9616; Nat. Commun., 2023, 14, 1873). Therefore, the observation of only Ni-O vibrational modes in our work is consistent with the standard characterization practice for such multi-metallic systems and does not imply that other metals are inactive or absent. Their participation has been confirmed through complementary techniques such as XPS and XAS, as presented in the manuscript.

Secondly, for the observed reduction in the number of stacked layers, it is likely associated with the influence of Na^+ on the interlayer interactions and stacking mode. Unlike transition metal cations that are tightly coordinated within the LDH slabs, Na^+ is more loosely bound and can interact with interlayer species such as water molecules and hydroxyl groups through ionic or hydrogen bonding. These interactions may potentially disturb or rearrange the original interlayer hydrogen-bonding network and electrostatic environment, thereby altering the stacking sequence and possibly reducing the number of stable layers. This hypothesis is in line with previous studies showing that the incorporation of heteroatomic species into LDH structures, either by direct doping or post-synthetic modification, can affect the interlayer coupling and stacking periodicity, occasionally resulting in fewer stacked layers (Nat. Catal. 2021, 4, 1050; Nat. Commun. 2024, 15, 9012; Nat. Commun. 2019, 10, 1711). In particular, monolayer and few-layer LDH systems have been reported to exhibit high sensitivity to structural and electronic perturbations introduced by foreign ions, which could favor thinner stacking configurations or promote partial delamination under certain conditions. As such, the structural transition observed in our Na-HE LDH samples may reasonably be attributed to

these Na-induced modifications to the interlayer environment. We have included this discussion in the revised manuscript to clarify the potential origin of the layer number change.

Actions:

- **Action 1, on Page 6, Line 134 of the revised manuscript**

Added discussion:

“This reduction in layer number may be related to the presence of Na ions, which could potentially modulate the interlayer interactions and stacking behavior by affecting the hydrogen-bonding network or local electrostatic environment.”

Reviewer’s comment (3):

The M-O covalency is strongly correlated to the LOM, while in this manuscript, the O K-edge has not been characterized and the variation of the M-O covalency has also not been discussed in deep. Moreover, the electronic configuration/chemical state differences between Na-HE LDH and HE LDH have not been reasonably explained.

Answer:

*We sincerely thank you for these insightful comments. To address the first point, we have added O K-edge soft X-ray absorption spectroscopy (sXAS) measurements (**Fig. S13**) and the corresponding discussion in the revised manuscript. The O K-edge spectra, collected in total electron yield (TEY) mode, exhibit distinct pre-edge features at ~534.5 eV and ~540.5 eV, corresponding to O 2p–M 3d and O 2p–Ni 4sp hybridization, respectively. Compared with HE LDH, Na-HE LDH shows a stronger pre-edge peak and a slight shift toward lower energy, which are indicative of enhanced M–O orbital hybridization and a higher 3d transition metal valence state. These results suggest increased M–O covalency, which is generally associated with greater lattice oxygen activity and thus supports the activation of the LOM pathway. Regarding the second point on electronic configuration and chemical state differences, we have expanded the discussion in multiple sections of the revised manuscript. First, high-resolution XPS analysis (**Fig. S10**) shows that the binding energies of 3d transition metals shift to higher values upon Na doping, suggesting an increase in their oxidation states. This increase implies a redistribution of d-electron occupancy, which may strengthen the M–O covalency by enhancing orbital overlap between metal 3d and oxygen 2p orbitals.*

*Second, we have incorporated density of states (DOS, **Fig. 6a**) analysis comparing Na-HEOOH and HEOOH. The DOS results show a significantly enhanced overlap between the O 2p and M 3d states in Na-HEOOH, reflecting stronger orbital hybridization. This increased overlap not only confirms the electronic structure modulation induced by Na incorporation, but also further supports the enhancement of M–O covalency and its correlation with lattice oxygen redox activity.*

Taken together, the combined evidence from O K-edge sXAS, XPS, and DFT calculations provides a consistent and coherent explanation of how Na doping modulates the electronic configuration and chemical states of the system.

Actions:

- **Action 1, on Page 7, Line 168 of the revised manuscript**

Added discussion:

“To further reveal the changes in metal valence and M-O bond covalency, O K-edge soft X-ray absorption spectroscopy (sXAS) measurements of HE LDH and Na-HE LDH were carried out using the total electron yield (TEY) mode. sXAS spectra (**Fig. S13**) of the O K-edge of HE LDH and Na-HE LDH show distinct peaks at 534.5 and 540.5 eV, which are attributed to O 2p-M 3d hybridization and O 2p-Ni 4sp hybridization with, respectively. It is noteworthy that the shift of the pre-edge peak to lower energies in Na-HE LDH is associated with an increase in the valence of the 3d element. Moreover, Na-HE LDH exhibits a stronger pre-edge peak (534.5 eV) compared to HE LDH, which indicates enhanced metal-oxygen bond covalency.”

• **Action 2, on Page 18 of the Supplementary Information**

Added figure:

Figure S13. O K-edge soft X-ray absorption spectroscopies (sXAS) of Na-HE LDH and HE LDH.

• **Action 3, on Page 6, Line 154 of the revised manuscript**

Revised discussion:

“This observed increase in the chemical state upon Na incorporation suggests a redistribution of d-electron occupancy, which may strengthen the M–O covalency by enhancing orbital overlap between metal 3d and oxygen 2p orbitals. This electronic

modulation is expected to facilitate lattice oxygen participation, thereby favoring the LOM pathway over the conventional AEM during the OER process.”

• **Action 4, on Page 17, Line 425 of the revised manuscript**

Added discussion:

“Moreover, we observed a notably enhanced overlap between the O 2p and M 3d states in Na-HEOOH compared to HEOOH, reflecting a stronger orbital hybridization. This increased overlap is indicative of enhanced M–O covalency, which is beneficial for promoting electron delocalization and is generally correlated with enhanced lattice oxygen activity.”

Reviewer’s comment (4):

Na-HE LDH shows a higher charge transfer rate and a better hydrophilic ability than HE LDH, while the authors have not discussed these differences. There are many similar structural description deficiencies in the manuscript, the authors need to rigorously revise their manuscript.

Answer:

*Thank you for pointing out the lack of discussion. We have now revised the manuscript to include detailed analyses from structural characterization, performance evaluation, and theoretical calculations to address these aspects in a more comprehensive and rigorous manner. For the charge transfer rate, from the perspective of electronic structure, the density of states (DOS) near the Fermi level (**Fig. S44**) shows that Na doping leads to a broader and more continuous distribution of states, with a higher DOS magnitude at the Fermi level compared to HE LDH (Nat Commun., 2023, 14, 997; Appl. Catal. B, 2024, 348, 123830). This implies enhanced electronic conductivity and facilitates more efficient electron transport through the material, theoretically supporting the observed lower charge transfer resistance from EIS measurements.*

*Regarding the hydrophilic ability, we again attribute the improved wettability of Na-HE LDH to its roughened nanosheet morphology, which increases surface energy and provides capillary-like structures that promote water spreading and adsorption. In addition, we have newly added DFT calculations of water adsorption energies (**Fig. S23**), which reveal that the water adsorption energy on Na-HE LDH is significantly lower (-0.836 eV) than that on pristine HE LDH (-0.177 eV). This finding is consistent with the smaller contact angle observed in the previous experiment (**Fig. S22**), confirming the enhanced hydrophilicity from a theoretical standpoint.*

All the above discussions have now been included in the main text, ensuring that the superior charge transfer kinetics and hydrophilic behavior of Na-HE LDH are cross-validated by experimental observation, theoretical modeling, and electrochemical performance. Furthermore, we have carefully reviewed the entire manuscript and revised similar descriptions with deficiencies to ensure that all structural and mechanistic interpretations are discussed with sufficient depth and scientific rigor.

Actions:

- **Action 1, on Page 9, Line 228 of the revised manuscript**

Added discussion:

“The improved wetting properties of Na-HE LDH may be attributed to its rough nanosheet morphology, which promotes water adsorption. This was confirmed by the calculated water adsorption energies (**Fig. S23**), which were significantly lower on Na-HE LDH (-0.836 eV) than on pristine HE LDH (-0.177 eV), as well as thermodynamically better hydrophilicity.”

- **Action 2, on Page 17, Line 416 of the revised manuscript**

Added discussion:

“We first calculated the total density of states (TDOS) (**Fig. S44**) and found that the DOS near E_f becomes more continuous and intensified after Na doping, indicating enhanced electronic conductivity, which aligns well with the improved charge transfer behavior observed in EIS measurements.”

Figure S22. Contact angle of liquid drop on catalyst surface.

- **Action 3, on Page 29 of the Supplementary Information**

Added figure:

Figure S23. Calculated adsorption free energy diagrams on the HE LDH (Ni as active site) and Na-HE LDH (O as active site).

- **Action 4, on Page 53 of the Supplementary Information**

Added figure:

Figure S44. Total density of states of Na-HEOOH and HEOOH.

Reviewer's comment (5):

Post-stability-test material characterization (e.g., structural integrity, surface composition) and quantitative metal leaching data should be supplemented to substantiate the stability conclusion.

Answer:

*Thank you for your suggestions. We have given characterization results such as XRD, TEM, Mapping, EPR and ion leaching after stability testing in the Supplementary Information. The results show that Na-HE LDH remains hexagonal in structure (**Fig. S30**) and still has a nanosheet morphology (**Fig. S31**) after stability testing, and HRTEM shows an amorphous*

layer on the nanosheet surface (**Fig. S32**). Combined with the uniform distribution of Mn, Fe, Co, Ni, Cu and Na elements in the mapping (**Fig. S33**), it is confirmed that the amorphous layer is Na-HEOOH, which is an inevitable result of the metal oxide/hydroxide OER. EPR (**Fig. S35**) shows that the O vacancies generated by the in situ of the Na-HE LDH are still retained after the stability test. More importantly, compared with the conventional NiFe LDH (28.9%), the leaching rate of Fe ions in HE LDH (3.2%) and Na-HE LDH (1.1%) was greatly reduced and Na-HE LDH had the lowest leaching rate of Fe ions (**Fig. S36**), which further confirmed the robust stability of Na-HE LDH.

Actions:

- **Action 1, on Page 13, Line 340 of the revised manuscript**

Added discussion:

“The results show that Na-HE LDH still maintains a crystalline hexagonal hydroxalcalite structure (**Fig. S30**) and still has a nanosheet morphology (**Fig. S31**), but do not exclude the presence of a surface metallic amorphous layer. HRTEM (**Fig. S32**) observed an amorphous layer of about 20 nm, and the mapping (**Fig. S33**) results showed that the Mn, Fe, Co, Ni, Cu and Na elements were still uniformly distributed without obvious aggregation and segregation. In addition, the XPS results (**Fig. S34**) confirm that elements such as Mn, Fe, Co, Ni, Cu and Na are still present. while the EPR results (**Fig. S35**) confirm the retention of the Vo. More importantly, compared with the conventional NiFe LDH (28.9%), the leaching rate of Fe ions in HE LDH (3.2%) and Na-HE LDH (1.1%) was greatly reduced and Na-HE LDH had the lowest leaching rate of Fe ions (**Fig. S36**), which further confirmed the robust stability of Na-HE LDH.”

Fig. S30 XRD patterns of Na-HE LDH after long-term OER test.

Fig. S31 TEM image of Na-HE LDH after long-term OER test.

Fig. S32 HRTEM image of Na-HE LDH after long-term OER test.

Fig. S33 EDS-Mapping of Na-HE LDH after long-term OER test.

Fig. S35 EPR results of Na-HE LDH after long-term OER test.

Fig. S36 Leaching of Fe ions of NiFe LDH, HE LDH and Na-HE LDH after long-term OER test.

Reviewer's comment (6):

The current DFT calculations appear oversimplified, as they fail to adequately capture the complex multi-metal synergistic effects inherent to high-entropy systems.

Answer:

*Thank you for this insightful comment. We have conducted additional theoretical analyses to more comprehensively address this concern. Specifically, we constructed two comparative models: a high-entropy structure (random distribution of multiple metals) and a non-high-entropy structure (ordered arrangement of the same metal species). The schematic diagrams of the two models have been provided in the newly added **Fig. S49**. We then calculated the formation energies for both structures (**Fig. S50**). The results show that the high-entropy structure exhibits a significantly lower formation energy (-1.59 eV) compared to the non-high-entropy structure (-0.41 eV), confirming its greater thermodynamic stability. Furthermore, we evaluated the OER energy barriers along the AEM pathway (**Fig. S51**), revealing that the high-entropy structure has a lower energy barrier (1.29 eV) than the non-high-entropy structure (1.50 eV), demonstrating enhanced intrinsic OER activity. These results clearly indicate that the multi-metal synergistic effects intrinsic to the high-entropy configuration indeed contribute to improved stability and activity, which further supports the superiority of our designed system.*

Meanwhile, we would like to clarify that in our initial DFT calculations, the primary objective was to elucidate the local electronic structure modulation induced by Na doping and its role in facilitating the O_{NB} formation and LOM activation. Due to the intrinsic complexity of high-entropy systems, fully capturing the extensive configurational disorder and multi-metal cooperative interactions through first-principles calculations remains a major challenge in the field. Thus, adopting simplified local models to approximate high-entropy materials has become a common and widely accepted strategy in recent computational studies (Nat. Commun., 2025, 16, 3327; Nat. Commun., 2025, 16, 1037; Nat. Commun., 2024, 15, 6669; Nat. Commun., 2023, 14, 5936). These studies typically select representative local atomic configurations or perform limited random mixing to investigate essential features such as stability, electronic structure, and catalytic activity, instead of fully resolving the vast configurational space of high-entropy materials, which remains computationally prohibitive. As such, our modeling approach is consistent with current best practices.

Actions:

- **Action 1, on Page 19, Line 473 of the revised manuscript**

Added discussion:

“To elucidate the effect of high entropy on the activity and stability of OER from the perspective of theoretical study, we constructed a high-entropy model with disordered metal arrangement and a non-high-entropy model with ordered metal arrangement, as shown in **Fig. S49**. We calculated the formation energies of the two structures (**Fig. S50**). The results show that the formation energy of the high-entropy structure (-1.59 eV) is significantly lower compared to that of the non-high-entropy structure (-0.41 eV), which confirms that the high-entropy structure has higher thermodynamic stability. In addition, we also evaluated the OER energy barrier on the AEM pathway (**Fig. S51**) and found that the energy barrier of the high-entropy structure (1.29 eV) is lower than that of the non-high-entropy structure (1.50 eV), which suggests that its intrinsic OER activity is enhanced. These results clearly show that the polymetallic synergistic effect inherent in the high-entropy structure indeed contributes to the enhancement of stability and activity, which further proves the superiority of our designed system.”

- **Action 2, on Page 58 of the Supplementary Information**

Added figure:

Fig. S49 non-high-entropy model with ordered 3d transition metals and high-entropy model with disordered 3d transition metals (crimson: Mn, pink: Co, cyan: Fe, gray: Ni, purple: Na, red: O, white: H).

- **Action 3, on Page 59 of the Supplementary Information**

Added figure:

Fig. S50 The formation energy of high-entropy structure and non-high-entropy structure.

- **Action 4, on Page 60 of the Supplementary Information**

Added figure:

Fig. S51 Computed free energies (ΔG) of OER based on AEM of high-entropy structure and non-high-entropy structure.

- **Point by point response and action to the comments of Reviewer 4**

Reviewer's general comment:

In this manuscript, Wang and coauthors synthesized sodium-doped high-entropy hydroxide (Na-HE LDH) as an oxygen evolution reaction (OER) catalyst in an alkaline environment, demonstrating good performance and stability in both RDE and WE tests. The reviewer would like to specifically address the XAS and electrochemical test results and analysis.

Response and action:

Thank you for your pertinent comments. All the comments have been carefully addressed in the revised manuscript. All changes have been highlighted in red in the revised manuscript for your reviewing convenience.

Reviewer's comment (1):

In Figure 3d-f, the authors claim that the coordination number of Ni is lower in Na-HE LDH compared to pure HE LDH based on FT-EXAFS data and fitting. However, the reviewer has several concerns regarding this analysis. First, the FT-EXAFS data for Na-HE LDH and HE LDH (Figure 3d) exhibit a noticeably different pseudo shoulder peak (slightly below 1 Å), which suggests an improper or inconsistent selection of the Fourier transform window and range during data processing. To ensure the validity of the analysis, the reviewer requests that the authors provide the raw EXAFS data and k-weighted spectra in the Supplementary Information and conduct a more rigorous analysis to rule out potential artifacts. Second, in EXAFS fitting, extracting coordination information requires careful consideration, as the coordination number and Debye-Waller factor (which represents the mean square displacement of atoms in the lattice) are correlated. This concern is particularly relevant here, as the authors report different Debye-Waller factors for Na-HE LDH and HE LDH (1.7 vs. 2.1), suggesting that the fitting analysis may be problematic. The reviewer recommends analyzing the K-edge EXAFS of other metal species to determine whether the Debye-Waller factors of the two catalysts differ systematically. Additionally, the pseudo peak in the fitted curves appears significantly different, indicating that certain fitting parameters may have been set inconsistently. The reviewer urges the authors to carefully reassess their analysis to eliminate potential errors and ensure the reliability of their conclusions.

Answer:

*We appreciate the reviewer for the insightful comments, which have significantly improved the rigor of our EXAFS analysis. We have re-fitted the EXAFS data for Na-HE LDH and HE LDH while ensuring the same fitting conditions. All raw $\chi(k)$ data and k^2 -weighted spectra have been provided in new **Fig. R2**. In terms of Na-HE LDH and HE LDH, the data range used for data fitting in k -space (Δk) and R -space (ΔR) are 2-9 Å⁻¹ and 1.1-2.5 Å, respectively, which is emphasized in the notes to the new **Table S2**. All Debye-Waller factors values were strictly constrained to <0.01 Å² during fitting. Moreover, we have fitted the K-edge EXAFS for the other metal species (i.e., NiO) following the reviewers' comment. The Debye-Waller factor for the NiO is similar to those of Na-HE LDH and HE LDH, which justifies the reasonableness and accuracy of the re-fitting. It should be noted here that reviewers may see that the raw $\chi(k)$ data in **Fig. R2f** do not exactly match the fitted curves, and this is due to the fact that we have*

only fitted the first shell layer of NiO. The new fitted curves are updated in **Figs. 3d-3f** and **Table S2** of the revised manuscript.

Fig. R2 EXAFS fitting analysis for sample. FT-EXAFS fitting curve of (a) Na-HE LDH, (c) HE LDH and (e) NiO at Ni k-edge. Comparison between experimental and theoretical EXAFS signal of (b) Na-HE LDH, (d) HE LDH and (f) NiO at Ni k-edge.

Actions:

- **Action 1, on Page 7, Line 179 of the revised manuscript**

Revised figure:

Fig. 3 (d) Ni k-edge FT-EXAFS spectra of Na-HE LDH, HE LDH, Ni foil and NiO. FT-EXAFS fitting curve of (e) HE LDH and (f) Na-HE LDH at Ni k-edge, the insert showing the fitting model.

- **Action 2, on Page 62 of the Supplementary Information**

Added table:

Table S2. Fitting result of FT-EXAFS curves.

Bond	N ^a	R ^b (Å)	σ^{2c} (Å ²)	ΔE_0^d (eV)	R factor ^e
------	----------------	--------------------	---------------------------------	---------------------	-----------------------

	type					
NiO	Ni-O	6.00	2.06	0.0049	3.50	0.0184
Na-HE LDH	Ni-O	5.08	2.04	0.0054	-2.06	0.0075
HE LDH	Ni-O	6.06	2.04	0.0094	-1.80	0.0113

^a Coordination numbers. ^b Bond distance. ^c Debye-Waller factors. ^d Inner potential correction.

^e Goodness of fit. In term of NiO, the data range used for data fitting in k-space (Δk) and R-space (ΔR) are 2-10.3 \AA^{-1} and 1-2.1 \AA , respectively. In terms of Na-HE LDH and HE LDH, the data range used for data fitting in k-space (Δk) and R-space (ΔR) are 2-9 \AA^{-1} and 1.1-2.5 \AA , respectively.

Reviewer's comment (2):

When extracting metal oxidation states from edge energy, an important assumption is that the metal has a similar coordination environment across samples. Therefore, metallic compounds are not suitable as standards for hydroxides/oxides. The reviewer recommends using metal oxides and hydroxides with different oxidation states as standards.

Answer:

Thank you for your suggestion. Indeed, as you said, it is more appropriate to compare oxidation states with samples having the same coordination environment. We compared the XAFS curves (**Fig. S11**) of individual metals in Na-HE LDH and HE LDH with the corresponding metal oxides and re-fitted the valence states. As shown in **Fig. S12**, the calculated valence states of Mn, Fe, Co, Ni and Cu in Na-HE LDH are +2.49, +2.67, +2.42, +2.77 and +3.22, respectively, which are higher than the valence states of Mn (+2.38), Fe (+2.57), Co (+2.29), Ni (+2.71) and Cu (+2.81) in HE LDH, which is consistent with the results of XPS.

Actions:

- **Action 1, on Page 7, Line 164 of the revised manuscript**

Revised discussion:

“Based on linear extrapolation (**Fig. S12**), the valence states of Mn, Fe, Co, Ni and Cu in Na-HE LDH are calculated to be +2.49, +2.67, +2.42, +2.77 and +3.22, which is higher than that of Mn (+2.38), Fe (+2.57), Co (+2.29), Ni (+2.71), Cu (+2.81) in HE LDH, which is in agreement with the XPS results.”

- **Action 2, on Page 16 of the Supplementary Information**

Revised figure:

Fig. S11 Normalized K -edge XAFS spectra of (a) Mn, (b) Fe, (c) Co, (d) Ni and (e) Cu elements. Compared with HE LDH, all the absorption edges of Mn, Fe, Co, Ni and Cu in Na-HE LDH shift to higher energies, implying an increased valence of 3d transition metals.

• **Action 3, on Page 17 of the Supplementary Information**

Revised figure:

Figure S12 Correlation between the metal oxidation states and the energy position of the XANES spectra.

Reviewer's comment (3):

The XAS white line intensity is influenced by both the oxidation state and the compactness of orbitals. Therefore, using white line intensity alone to determine changes in oxidation state is not highly accurate. The reviewer recommends presenting and analyzing the entire XANES spectrum for a more reliable interpretation.

Answer:

*Thanks to your reminder, we have shown the entire XANES spectra of all elements (**Figs. 5f, 5i and Fig. S37-S38**), and in addition to analyzing the changes in the white line peaks, we have given the specific valence states of all the elements in HE LDH and Na-HE LDH at different applied voltages by linear extrapolation (**Fig. S39-S40**), where the valence states of all metal elements increase with increasing oxidation potential, indicating that HE LDH and Na-HE LDH are reconstruction. We note that Ni has the largest change in valence state among all elements in HE LDH (0.45, from +2.71 to +3.16) and in Na-HE LDH (0.48, from +2.77 to +3.25). This suggests that nickel is involved in the OER process through electron transfer and is evaluated as the active site. It should be noted here that although the LOM mechanism uses O as the active site, there is also a portion of the AEM mechanism throughout the OER, and Ni is the active site in the AEM mechanism, which leads to a drastic change in Ni valence. It is noteworthy that the higher valence of Ni in Na-HE LDH compared to HE LDH is more favorable to the OER process, which is consistent with the in-situ Raman results.*

Actions

- **Action 1, on Page 15, Line 380 of the revised manuscript**

Revised discussion:

“The in situ XANES spectra show that the valence states of Ni in Na-HE LDH and HE LDH increase significantly from the open-circuit potential (OCV) to 1.7 V (**Fig. 5g and Fig. 5j**) whereas the valence states of Mn, Fe, Co, Ni and Cu increased slightly (**Fig. S37-S38**). We further give the values of valence changes with increasing voltage for all metallic elements (**Figs. S39-S40**), where the valence of all the metal elements increases with increasing oxidation potential, suggesting that the HE LDH and Na-HE LDH are reconstruction. It is noted that among all the elements of HE LDH, the Ni has the largest change in valence state (0.45, +2.71 to +3.16), and the Ni also has the largest change in valence in Na-HE LDH (0.48, from +2.77 to +3.25). This indicates that Ni involved in the OER process through electron transfer and is evaluated as the active site. It is important to note here that although the LOM mechanism is O as the active site, there is also a part of the AEM mechanism in the whole OER, and Ni is the active site in the AEM mechanism, which leads to the drastic change of Ni valence. It is noteworthy that the valence state of Ni is higher in Na-HE LDH compared to HE LDH, which is more favorable to the OER

process, consistent with the in-situ Raman results.”

- **Action 2, on Page 46 of the Supplementary Information**

Revised figure:

Figure S37. In-situ Mn, Fe, Co and Cu k-edge XANES spectra of HE LDH under applied potentials of 1.1–1.7 V vs. RHE in 1.0 M KOH.

- **Action 3, on Page 47 of the Supplementary Information**

Revised figure:

Figure S38. In-situ Mn, Fe, Co and Cu k-edge XANES spectra of Na-HE LDH under applied potentials of 1.1–1.7 V vs. RHE in 1.0 M KOH.

- **Action 4, on Page 48 of the Supplementary Information**

Added figure:

Figure S39. Variation of metal oxidation state with voltage in HE LDH under applied potentials of 1.1–1.7 V vs. RHE in 1.0 M KOH.

- **Action 5, on Page 49 of the Supplementary Information**

Added figure:

Figure S40. Variation of metal oxidation state with voltage in Na-HE LDH under applied potentials of 1.1–1.7 V vs. RHE in 1.0 M KOH.

Reviewer's comment (4):

The in situ EXAFS data reveal significant changes in M–O distance, intensity, and the appearance/disappearance of the M–M peak (between 2–3 Å) (Figure 5h,k; Figure S27, S28), suggesting a change in oxidation state and structural transformation in the catalysts, likely from hydroxide to oxide/oxyhydroxide. Do the authors have any insights into this transformation and its reversibility?

Answer:

Thanks for your insightful comments. Indeed, the in-situ EXAFS data show a change in M–O bond distances and M–M coordination, which is consistent with a transition from an initially layered hydroxide structure to a high-valent oxygen hydroxide. This transition is further confirmed by our in-situ Raman and XANES analyses, in which we observe the progressive oxidation of metal species as the applied potential increases (Figs. 5a–5e, 5g and 5j). We summarize the significant changes in the M–O (~1.5 Å) distance in Fig. R3. For Na-HE LDH, it can be observed that the M–O distances for all elements shorten with increasing oxidation potential, indicating a structural transition from metal hydroxide to metal-oxygen hydroxide (Nature communications, 2025, 16, 726.). The M–M strength, on the other hand, becomes weaker with increasing oxidation potential, which may be attributed to the fact that a large amount of OH/H₂O is enriched at the surface and embedded in the interlayer lattice of the reconstructed Na-HE LDH at high oxidation potential, which results in the widening of the lattice spacing manifesting itself as broadening (Adv. Mater., 2022, 34, 2108541; Angew. Chem., 2024, 136, e202402371). It is noted that there is no significant change in the M–O and M–M of HE-LDH, suggesting that HE-LDH is less reconstruction than Na-HE LDH.

Fig. R3 Variation of M-O bond distances in HE LDH and Na-HE LDH at different applied voltages.

Reviewer’s comment (5):

The reviewer recommends including details of the electrochemical and WE tests, such as catalyst loading, scan rate, rotation rate, and iR correction, in the manuscript for clarity.

Answer:

Thank you for your suggestions. We have added details of electrochemical tests and WE tests, including catalyst loading, scan rate, scan range and iR correction, etc., in the electrochemical measurements section of the revised manuscript.

Actions:

- **Action 1, on Page 20, Line 523 of the revised manuscript**

Revised discussion:

“4.3 Electrochemical measurements

The electrochemical characterization was performed using a CHI 660E electrochemical workstation, operating at ambient temperature with a three-electrode setup. The working electrode was the Na-HE LDH, HE LDH and IrO₂ etc. (electrode area are 1×1 cm², catalyst loading ~ 0.9 mg cm⁻²), while a carbon rod served as the counter electrode and Hg/HgO as the reference electrode. Samples were activated and stabilized by multiple cyclic voltammetry (CV) curves across a potential range of 0 ~ 0.8 V vs. Hg/HgO at 100 mV s⁻¹ and 20 mV s⁻¹ before Linear sweep voltammetry (LSV) curves measure. LSV curve was

performed in a 1.0 M KOH solution, scanning at 2 mV s^{-1} across a potential range of $0.8 \sim 0 \text{ V vs. Hg/HgO}$ with 95% iR compensation. The potential was converted to the reversible hydrogen electrode (RHE) scale using the Nernst equation: $E_{\text{RHE}} = E_{\text{Hg/HgO}} + 0.059 \times \text{pH} + 0.095$. Electrochemical impedance spectroscopy (EIS) was conducted between $0.01 \sim 100 \text{ kHz}$ at 0.6 V vs. Hg/HgO . Cyclic voltammetry (CV) was used to measure double-layer capacitance (C_{dl}) and estimate the electrochemically active surface area (ECSA, $\text{ECSA} = C_{\text{dl}}/C_s$), with scan rates ranging from 5 to 100 mV s^{-1} within a potential window of $0.14 \sim 0.24 \text{ V vs. Hg/HgO}$. A standard specific capacitance (C_s) of $40 \mu\text{F cm}^{-2}$ was applied for these calculations.”

Reviewer’s comment (6):

Could the authors clarify why the AEMWE electrolyzer was tested in concentrated KOH (30%) instead of 1 M KOH or lower, given that lower KOH concentrations are less corrosive and a key focus in AEMWE research?

Answer:

Thank you for this valuable comment. We fully acknowledge that lower concentrations of KOH, such as 1 M, are less corrosive and have gained increasing attention in recent AEMWE research, particularly in relation to membrane stability and long-term system durability.

Nonetheless, in this study, we chose to use 30 wt% KOH to benchmark the catalyst under industrially relevant and practically demanding conditions. High-concentration alkaline electrolytes remain widely employed in commercial alkaline water electrolyzers and have been extended to AEMWE systems for evaluating catalytic performance and robustness. For instance, a recent study demonstrated stable AEMWE operation at 1 A cm^{-2} for 300 h in 30 wt% KOH at $80 \text{ }^\circ\text{C}$ using a Ru@Cu-based cathode (Nat. Commun. 2023, 14, 4680). Similarly, 3 M KOH (~16 wt%) was used in another full-cell study of Y-doped NiMo-MoO₂ electrocatalysts to achieve long-term durability (Nat. Commun. 2025, 16, 773).

Moreover, concentrated KOH solutions (typically 20-30 wt%) have long been utilized in large-scale alkaline electrolysis since the 1950s and continue to serve as performance benchmarks in the field (Chem. Rev. 2022, 122, 11830-11895; Chem. Soc. Rev. 2022, 51, 9620-9693). As such, our choice of 30 wt% KOH thus reflects a deliberate attempt to assess not only the intrinsic activity but also the practical stability of Na-HE LDH under rigorous and widely adopted operation conditions.

School of Materials Science and Engineering

Tel: +86-0533-2788087

Shandong University of Technology

Fax: +86-0533-2788087

Zibo 255000, P. R. China

E-mail: conghailin@sdut.edu.cn (H. L. Cong)

Jun. 3, 2025

Title: Constructing oxygen non-bonding band in high-entropy (oxygen) hydroxides for industrial-scale water oxidation

Dear Editors and Reviewers:

We appreciate your time and efforts in handling our manuscript in *Nature Communications*. The manuscript has been revised and greatly improved according to reviewers' suggestions and comments. Responses to reviewers' comments are listed as follows.

Thank you very much for your further consideration.

Sincerely yours,

Dr. Hailin Cong

Reviewer #1 (Remarks to the Author): I appreciate the authors' thorough and careful revision of the manuscript. They have addressed the previous concerns effectively and improved the clarity and quality of the presentation. The additional data and explanations were effective and improved the clarity and quality of the presentation. The additional data and explanations provided have strengthened the arguments and enhanced the overall impact of the study. I have a few additional comments.

1. There is a significant peak shift in the overall O 1s spectra between HE LDH and Na HE LDH. It would be helpful if the authors could verify this observation and provide a brief explanation.

Answer: After Na doping, the XPS spectra of O 1s are shifted towards lower binding energies, while all transition M 2p orbitals are shifted towards higher binding energies, indicating that the introduction of Na induces strong electronic interactions between the transition metal and oxygen. Moreover, the negative shift of the O 1s binding energy and the positive shift of the M 2p binding energy indicate an enhancement of the covalency of the M-O bond in favor of the LOM mechanism (*Nature Communications*, (2024) 15:2501; *Energy Environ. Sci.*, 2024, 17, 5260-5272).

Changes in the revised manuscript

Added discussion: After Na doping, the XPS spectra of O 1s are shifted towards lower binding energies, while all transition M 2p orbitals are shifted towards higher binding energies, indicating that the introduction of Na induces strong electronic interactions between the transition metal and oxygen. Moreover, the negative shift of the O 1s binding energy and the positive shift of the M 2p binding energy indicate an enhancement of the covalency of the M-O bond in favour of the LOM mechanism

2. I suggest the authors provide the applied potential/voltage for all chronoamperometry stability graphs.

Answer: Thanks for your suggestion, we have provided applied potentials in the figure notes of all chronoamperometry stability graphs in the revised manuscript and supporting information (**Fig. 4k**, **Fig. S26**, **Fig. S27**, **Fig. S28e** and **Fig. S30c**).

3. In the comparison table, the authors should consider providing the complete composition instead of abbreviations such as HEG, HEA, MCPS, etc.

Answer: Thank you for the suggestions. We have removed the abbreviations from the

cross-reference tables and provided the full sample composition for the reader.

Table S4. Comparison of OER performances of Na-HE LDH with previously reported well-performed OER electrocatalysts.

Catalyst	$\eta@$ 10mA cm ² (mV)	$\eta@$ 50 mA cm ⁻² (mV)	$\eta@$ 100 mA cm ⁻² (mV)	Electrolyte	Ref
Na-MnFeCoNiCu LDH (Na-HE LDH)	176	228	263	1M KOH	This work
(Co, Cu, Fe, Mn, Ni) ₃ O ₄ /multi-walled carbon nanotubes	350	-	-	1M KOH	1
CV-activated MnFeCoNi HEA	302	-	-	1M KOH	2
FeCoNiCrNb _{0.5}	288	-	-	1M KOH	3
HF-treated CoFeNi	265	-	-	1M KOH	4
Fe-Cr-Co-Ni-Cu HE-LDHs-Ar-20	330	-	-	1M KOH	5
(CoNiMnZnFe) ₃ O _{3.2}	336	-	-	1M KOH	6
La ₅ (CrMnFeNi) ₂ Co	380	-	-	1M KOH	7
CoCrFeMnNi glycerate	229	-	-	1M KOH	8
NiCoFeCrMo-based HEH	292	-	-	1M KOH	9
FeCoNiMn	266	-	-	1M KOH	10
FeNiCoAl	225	-	-	1M KOH	11
CuCoNiFeMn high-entropy-alloy -60 h	-	479	-	1M KOH	12
V _{1.0} -CuCoNiFeMn High-Entropy Alloy	-	370	-	1M KOH	13

FeCoNiCuMn@CF HEAs		260	1M KOH	14
Au _{SA} -MnFeCoNiCu LDH	213	260	1M KOH	15

4. I suggest the authors add a comparison table for different Na doping levels, including activity, Tafel slope, ECSA, and EIS data.

Answer: Thank you for your suggestion. We have added a comparison table for different Na doping levels including activity, Tafel slope, ECSA and EIS data at **Table S1**.

Table S1. Compare activity, Tafel slope, ECSA and EIS for Na_{0.025}-HE LDH, Na_{0.045}-HE LDH and Na_{0.065}-HE LDH.

Catalyst	Activity (mV)	Tafel slope (mV dec ⁻¹)	ECSA (cm ²)	EIS (Ω)
Na _{0.025} -HE LDH	232	38.5	105	4.9
Na _{0.045} -HE LDH	176	29.6	62.5	2.8
Na _{0.065} -HE LDH	277	41.1	150	5.6

5. The authors could provide a high-magnification TEM image of the sample (before testing) to clarify the amorphous layer formation during the OER process.

Answer: Thanks to your suggestion, we have added the high magnification TEM image of the sample before testing in **Fig. S6** of the revised Supporting Information, and the presence of the amorphous layer was not observed from the image, confirming that the amorphous layer was formed after the OER test.

Fig. S6 High-resolution TEM images of Na-HE LDH.

Changes in the revised manuscript

Added discussion: It is also observed that the Na-HE LDH are all crystalline structures with no amorphous layer attached (**Fig. S6**).

6. I recommend that the authors use a line graph instead of scatter plots for the EPR data presentation.

Answer: Thank you for your suggestion. We now present the EPR data in line graphs, see **Fig. 3c** and **Fig. S36** in the revised manuscript and supporting information.

Fig. 3c EPR spectra of Na-HE LDH and HE LDH. Signal at $g = 2.005$ proves the presence of V_o in Na-HE LDH.

Fig. S36. EPR results of Na-HE LDH after long-term OER test.

7. Please include the electrolyte flow rates in the AEMWE section for better reproducibility and clarity.

Answer: Thank you for your suggestion. We have listed the flow rate of the 30 wt% KOH electrolyte (100 sccm) in the AEMWE assembly section to improve the reproducibility and clarity of the experiment.

Changes in the revised manuscript

Added discussion: The temperature of the electrolytic cell was maintained at 60 °C,

and 30 wt% KOH was used as the electrolyte. A peristaltic pump delivered the electrolyte into the electrode channel at a flow rate of 100 sccm.

Reviewer #2 (Remarks to the Author): The author has carefully revised the comments, and I agree to accept the revised paper.

Answer: Thank you very much.

Reviewer #3 (Remarks to the Author): The authors have addressed most of my concerns, and I am pleased to recommend acceptance of this manuscript in its current form.

Answer: Thank you very much.

Reviewer #4 (Remarks to the Author): After revision, the authors have adequately addressed the concerns previously raised, particularly those related to the processing and analysis of the XAS data. The additional details and clarifications provided in both the revised manuscript and the point-by-point response demonstrate a sound understanding of the methodology and ensure that the data analysis is both transparent and scientifically robust. Based on these revisions, the reviewer is satisfied that the XAS data processing and analysis now meet the standards required for publication in this journal

Answer: Thank you very much.